# Boron-induced transformation of ultrathin Au films into two-dimensional metallic nanostructures

Alexei Preobrajenski [1] ✉, Nikolay Vinogradov[1], David A. Duncan [2], Tien-Lin Lee[2], Mikhail Tsitsvero [3], Tetsuya Taketsugu [3,4] & Andrey Lyalin [4,5] ✉

The synthesis of large, freestanding, single-atom-thick two-dimensional (2D) metallic materials remains challenging due to the isotropic nature of metallic bonding. Here, we present a bottom-up approach for fabricating macroscopically large, nearly freestanding 2D gold (Au) monolayers, consisting of nanostructured patches. By forming Au monolayers on an Ir(111) substrate and embedding boron (B) atoms at the Au/Ir interface, we achieve suspended monoatomic Au sheets with hexagonal structures and triangular nanoscale patterns. Alternative patterns of periodic nanodots are observed in Au bilayers on the B/Ir(111) substrate. Using scanning tunneling microscopy, X-ray spectroscopies, and theoretical calculations, we reveal the role of buried B species in forming the nanostructured Au layers. Changes in the Au monolayer's band structure upon substrate decoupling indicate a transition from 3D to 2D metal bonding. The resulting Au films exhibit remarkable thermal stability, making them practical for studying the catalytic activity of 2D gold.

The vast majority of two-dimensional (2D) materials synthesized experimentally to date can be traced back to three-dimensional layered van der Waals materials, such as graphite, hexagonal BN or transition metal dichalcogenides. Strong directional covalent in-plane bonds and weak van der Waals interlayer interactions in these materials facilitate the exfoliation, wrinkling, folding, rolling, and other manipulations of individual layers. However, replacing the directional in-plane bonding with more delocalized types of interactions poses challenges for exfoliation and handling of individual 2D layers. A good example here is 2D boron or borophene films[1], where both directional and more delocalized three-centre bonds contribute to the in-plane interaction. Although there have been successful instances of liquid-phase exfoliation of few-layer-thick B flakes from bulk B powder[2], strictly single-layer free-standing borophene is hard to realize, as covalent B-B bonds tend to form between adjacent layers, resulting instead in ultrathin 3D B films[3]. It is reasonable to assume that materials with even stronger delocalization of in-plane interactions may not allow for the experimental separation of strictly single-atom-thick freestanding monolayers. In particular, elemental metals are typically not considered suitable for forming 2D materials due to the strongly delocalized metallic bonding, which tends to form isotropic structures. Thus, while it is straightforward to grow 2D metal films on a wetting substrate, envisioning an unsupported monoatomic metal layer as a tangible physical object is counterintuitive due to the essentially 3D nature of metallic bonding.

Nonetheless, the exotic electronic structure and properties of 2D metals have called for significant efforts towards experimental realization of various 2D metal structures[4]. For example, an ability to form small freestanding single-atom-thick Fe patches in the pores of graphene membranes has been demonstrated experimentally[5] and predicted theoretically for a variety of other metals[6]. Additional relevant examples include the synthesis of freestanding hexagonal 2D Pd

[1]MAX IV Laboratory, Lund University, 221 00 Lund, Sweden. [2]Diamond Light Source, Didcot OX11 0QX, UK. [3]Institute for Chemical Reaction Design and Discovery (WPI-ICReDD), Hokkaido University, Sapporo 001-0021, Japan. [4]Department of Chemistry, Faculty of Science, Hokkaido University, Sapporo 060-0810, Japan. [5]Research Center for Energy and Environmental Materials (GREEN), National Institute for Materials Science, Namiki 1-1, Tsukuba 305-0044, Japan. ✉e-mail: Alexei.Preobrajenski@maxiv.lu.se; lyalin@icredd.hokudai.ac.jp

nano-sheets with a thickness of several atomic layers[7], and a manu-facturing of flexible, ultrathin, and large area (>100 μm²) single-crystalline Au membranes with atomically flat surfaces[8]. Furthermore, the recent advancements in the synthesis of 2D noble metals have been summarized by I. Shtepliuk[9], who also reviewed prospects for their application in hydrogen evolution reaction catalysis. Still, whenever 2D metal structures can be produced, they are either very small, not precisely single-atom thick or not freestanding. In the quest for truly 2D metallic materials, a seemingly appealing strategy would be to grow metal monolayers on a suitable substrate and then elevate them by intercalating foreign species. Although this approach is known to work for covalently bonded 2D materials with their "blanket-like" structure, for the 2D metals it is difficult to realize, as any intercalant would randomly disrupt the delicate monolayer and interact with it. On the other hand, a direct growth of metal monolayers on an inert and atomically-flat substrate like graphene does not look promising either, because metals tend to agglomerate into 3D nanoparticles on such surfaces in attempt to reduce total surface energy. Although, at lower temperatures metallic monolayers may form even on inert substrates, like e.g. monoatomic potassium islands grown on graphite[10].

While experimental evidence for the synthesis of freestanding and truly single-layered metal sheets remains elusive, the properties of such systems can be predicted theoretically. In a systematic work by J. Nevalaita and P. Koskinen[11] as many as 45 elemental metals were studied as atomically thin films, each of them in three stable 2D lattice structures: hexagonal, square, and honeycomb. A clear linear correlation was established between 2D and 3D metals regarding their cohesive energies, equilibrium bond lengths, and bulk moduli. Also, for individual metals, the existence of 2D atomic layers was suggested theoretically, like for Au[12], Ag[13] and Cu[14], with all coinage metals predicted to stabilize as hexagonal close-packed monolayers. Therefore, in theory, it should be feasible to synthesize 2D metals that are (1) macroscopically large, (2) single-atom thick, and (3) freestanding. However, the main question remains unanswered: is it possible to fabricate such monolayers experimentally, with all three above-mentioned conditions satisfied?

For gold monolayers specifically, the formation of freestanding 2D Au membranes has been demonstrated in a top-down approach using transmission electron microscopy, either by mechanical thinning the bulk material[15] or by de-composing the bulk AuAg alloy[16], with the resulting membranes being nanometre-sized. Another method involves using Au intercalation to form a buried Au monolayer, like the one stabilized between silicon carbide and graphene[17]. In this case, the monolayer is truly two-dimensional and can be made macroscopically large due to the bottom-up approach to its fabrication. However, it can hardly be considered freestanding, as the distance of 3.08 Å[17] between neighbouring Au atoms matches the lattice parameter of the underlying SiC(0001), being significantly larger than the bulk Au-Au bond length of 2.88 Å. On the other hand, the 3D → 2D (hexagonal) transition in Au is expected to result in shrinking the lattice constant and the Au-Au bond length to 2.76 Å[11,12]. A different route for the production of 2D Au films can be offered by wet chemistry. For instance, a recent method for synthesizing Au 2D atomic sheets involves exchanging Si in layered $Ti_3SiC_2$ material with intercalated Au, followed by etching away $Ti_3C_2$[18]. Despite their advantages of simplicity and scalability, wet chemistry methods lack the precise control inherent in bottom-up on-surface synthesis.

In this study, we demonstrate the feasibility of growing a macroscopically large and nearly freestanding 2D Au monolayer, composed of periodic nanostructured patches. The proposed bottom-up approach involves a formation of the Au monolayer on the Ir(111) substrate followed by an ordered embedding of B atoms at the Au/Ir interface. This results in a regular nanostructured suspended monoatomic Au sheet with a hexagonal structure and a slightly modulated lattice constant. Furthermore, the structural transition from 2D to 3D is observed upon the formation of the second Au monolayer, leading to an Au bilayer with an alternative nanostructuring pattern. The role and properties of the buried B species are revealed by a combination of several experimental techniques and density functional theory (DFT) calculations. Thermal stability of the resulting Au films is significant, making them a convenient and practical platform for testing theories regarding catalytic activity of two-dimensional gold[19].

## Results and discussion

### Nanostructuring mono- and bilayer Au via B embedding

The first atomic monolayer (ML) of gold grows epitaxially on the Ir(111) substrate in a pseudomorphic manner (see the supplementary information (SI) for the growth process details), adopting the Ir lattice constant and causing no moiré patterns either in low-energy electron diffraction (LEED) or in scanning tunnelling microscopy (STM). However, upon exposure to the flux of elemental B at elevated temperatures, the surface morphology changes drastically: a very regular pattern of densely packed equilateral triangles starts to emerge on the (111) terraces (Fig. 1a). The arrays of these triangles usually propagate from the step edges but can also form in the middle of a terrace. At a B coverage approaching 1 ML (defined as one B atom per one Au atom in the closed-packed Au ML), the entire surface is transformed into a single domain of highly regular triangular nanostructures with very low density of defects (Fig. 1b) and a characteristic stripy LEED pattern (Fig. 1b, inset). A closer look at these structures (see Fig. 1c-d) shows that the adjacent triangles exhibit alternating sizes, resulting in the periodic supercell shaped like a rhombus, with one slightly larger and one slightly smaller triangle. From an analysis of the autocorrelation function (like the one shown in the inset of Fig. 1c) the lattice parameter of the rhombus supercell is found to be 11.7 ± 0.1 nm. The apparent height difference between the triangle surfaces and the separating dark trenches/corners is measured to be 0.9 ± 0.1 Å. The atomically resolved images, such as the one shown in Fig. 1d, reveal that the atoms on both triangles of the supercell are arranged in a close-packed fashion, with an interatomic distance of 2.8 ± 0.1 Å. This observation strongly suggests that the surface of the triangles constitutes a close-packed Au monolayer. The apparent absence of B atoms in the images implies that they do not reside on the surface but instead penetrate beneath the Au monolayer.

As boron has been reported to form borophene atop the Ir(111) surface[20], it is plausible to assume that the B atoms accumulate at the Au/Ir interface, binding to the Ir substrate. Indeed, the formation of B-Ir bond can be clearly traced in the high-resolution surface-sensitive Ir $4f$ X-ray photoelectron spectroscopy (XPS) signals (Fig. 1e-g), which probe the topmost 2-3 ML of the substrate. In the case of clean Ir(111), the Ir $4f_{7/2}$ spectrum is split into the bulk- and surface-related components Ir1 and Ir2 at the binding energies (BE) of 60.85 eV and 60.35 eV, respectively (Fig. 1e). Note that the uncertainties in the reported BEs are consistently below 0.02 eV for all core levels discussed in this study. Once a ML of Au is formed directly on Ir(111), the surface-related component Ir2 turns into an interface-related component Ir3 at a BE of 60.45 eV (Fig. 1f). After adding 1 ML of B to this system at 550°C, a new interface-related component Ir4 emerges at a BE of 60.63 eV (Fig. 1g), indicating the formation of B-Ir bonds with the topmost Ir layer. Indeed, a reduction in surface core-level shifts caused by adsorbates is known to be a sign of chemisorption and the formation of a chemical bond[21]. The significant BE shift of 0.28 eV between Ir2 and Ir4 suggests that this bonding is rather strong, with a considerable electron transfer from Ir to B. The Ir-B bond is a typical case of the metal-nonmetal interaction, as metalloid boron tends to behave more like a nonmetal in terms of its chemical properties. Metals, particularly transition metals, tend to donate electrons when bonding with nonmetals. Therefore, it is not unusual to assume an electron transfer from Ir to B in our system. The survival of the Ir3 component in

Fig. 1g implies that certain regions of the Au ML maintain direct contact with Ir(111).

Another intriguing case involves a 2 ML thick Au film. On the pristine Ir(111) surface it forms a characteristic hexagonal moiré pattern (see Fig. 2d and Supplementary Figs. 1b and 3c) with a supercell lattice parameter of $7.4 \pm 0.2$ nm. This suggests a partial relaxation of the interatomic distance, d(Au-Au), from the pseudomorphic value of the first Au monolayer (2.72 Å) to a larger value of 2.82 Å in the second layer, while still being smaller than the interatomic distance in bulk Au (2.88 Å). The morphology of this Au film changes drastically upon exposing it to a flux of elemental B at elevated temperatures. As can be seen in Fig. 2a–c, the surface becomes uniformly covered with periodically arranged nano-dots. On the samples where the Au coverage is between 1 and 2 ML, the nano-pattering characteristic for these two thicknesses can coexist, giving rise to areas with triangles and dots, respectively (see Fig. 2a). In the large-scale image depicted in Fig. 2b, one can see that the periodicity of the dot arrangement is somewhat less regular than the triangular arrangement (compare with Fig. 1b). However, moiré spots are now clearly visible in LEED (Fig. 2b, inset), reflecting a smaller size of the supercell. The dot shapes and sizes are similar but not identical (see Fig. 2c), and their apparent height measures $0.95 \pm 0.15$ Å. By using the autocorrelation function on several images (such as the one in the inset of Fig. 2c), the supercell parameter is found to be $6.7 \pm 0.3$ nm. Upon closer inspection with atomic resolution (see Fig. 2c, inset), both the surface of the dots and of the inter-dot spacing show a characteristic closed-packed pattern of the Au

atoms, similar to the triangular nanostructures shown in Fig. 1. In Fig. 2d, the morphology of a pristine (without boron) 2 ML thick Au film is shown for comparison, using the same colour (height) scale as in Fig. 2c. Unlike the B-induced dotted nano-pattern seen in Fig. 2a–c, it exhibits only marginal apparent height variation below 0.2 Å. Typical apparent height profiles across images in Figs. 1c and 2c-d can be found in the Supplementary Information (Supplementary Fig. 3).

In addition to STM and LEED, XPS can independently confirm that it is indeed one and two Au monolayers, which break into the triangular and dotted nano-patterns, respectively. The most obvious indicator here is the Au $4f$ photoelectron spectrum obtained in the most surface-sensitive mode. As can be seen in Fig. 2e, the Au $4f_{7/2}$ spectrum from a single Au monolayer on B/Ir(111) reveals only one component with a BE of 83.64 eV. In contrast, the same spectrum from the two ML thick Au layer on B/Ir(111) shows a second component at 83.94 eV, while the original (surface-related) component remains unchanged. This second component originates from the fully coordinated (bulk-like) Au atoms of the bottom layer, as its energy is very close to 84.0 eV – the binding energy of Au $4f$ electrons in the bulk. The intensity ratio of these two components is 0.55 in favour of the surface component, matching the ratio expected for exactly 2 ML thick Au film. Indeed, the inelastic mean free path of electrons with a kinetic energy of 80 eV in bulk gold can be estimated as 3.9 Å using the method by Shinotsuka et al.[22]. Given the interlayer distance of 2.35 Å for Au(111) atomic planes, the intensity of the Au $4f$ component from the second (lower lying) Au layer has to be 55% of the intensity from the top Au layer. This is

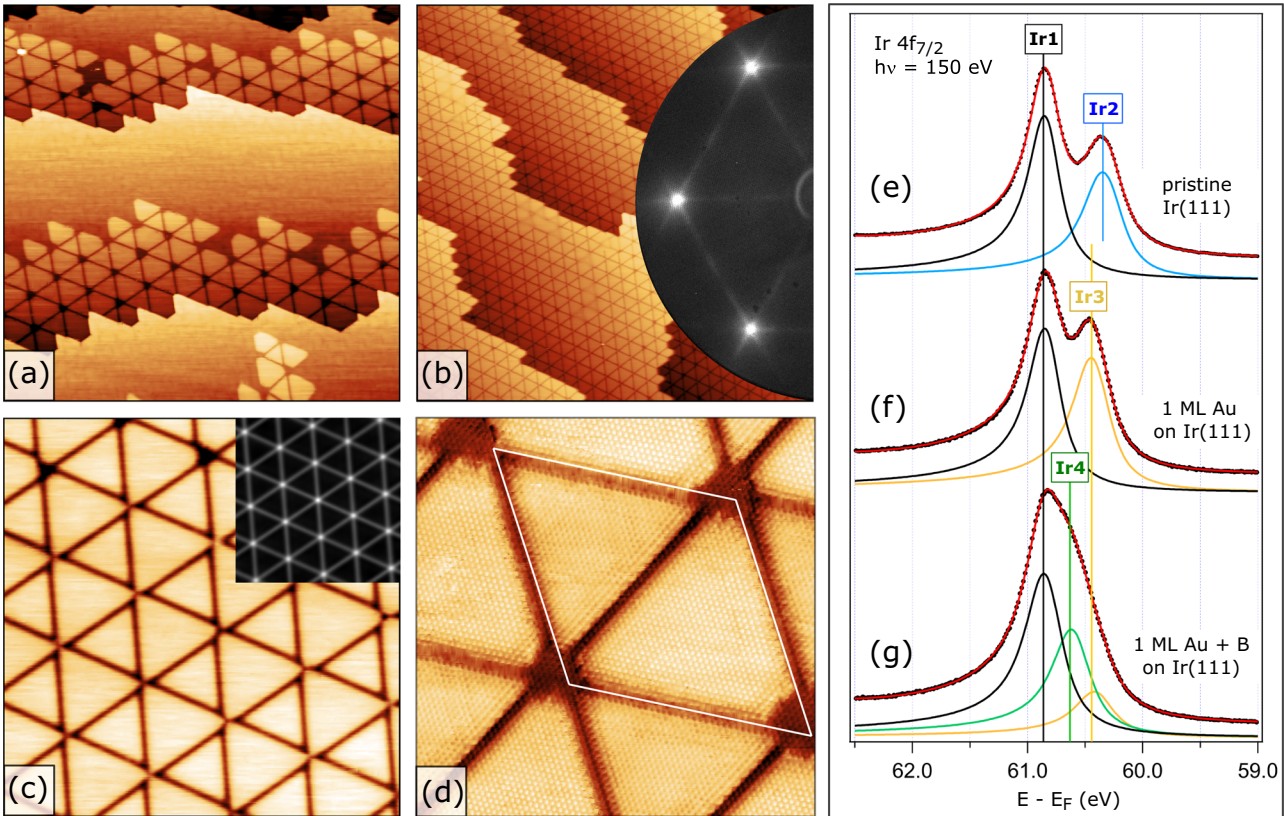

**Fig. 1 | Nano-structuring of a single-atom thick Au layer on Ir(111) by embedding interfacial B. a** 150 nm scanning tunnelling microscopy (STM) image of 1 monolayer (ML) Au / Ir(111) after depositing ca. 0.5 ML B at 550 °C, (**b**) 300 nm STM image of 1 ML Au / Ir(111) after depositing ca. 1 ML B at 550 °C showing long-range ordering, inset: low-energy electron diffraction (LEED) pattern from this surface taken at E = 70 eV, (**c**) same as (**b**) but 50 nm image, inset: auto-correlation function from this image, (**d**) atomically-resolved close-up STM image of the nanostructured Au monolayer with the unit supercell indicated by a white rhombus, (**e-f**) Ir $4f_{7/2}$ photoelectron spectra taken with the photon energy (hv) of 150 eV from (**e**) the clean Ir(111) surface, (**f**) ca. 1 ML Au on Ir(111), (**g**) ca. 1 ML Au on Ir(111) nano-structured by B incorporation at the Au/Ir interface. The binding energy scale is referred to the Fermi level ($E_F$). Black dots represent experimentally measured data points; red curve is a result of the peak fit analysis. Black, blue, orange and green curves represent peak fit components denoted as Ir1, Ir2, Ir3 and Ir4, respectively, as described in the text.

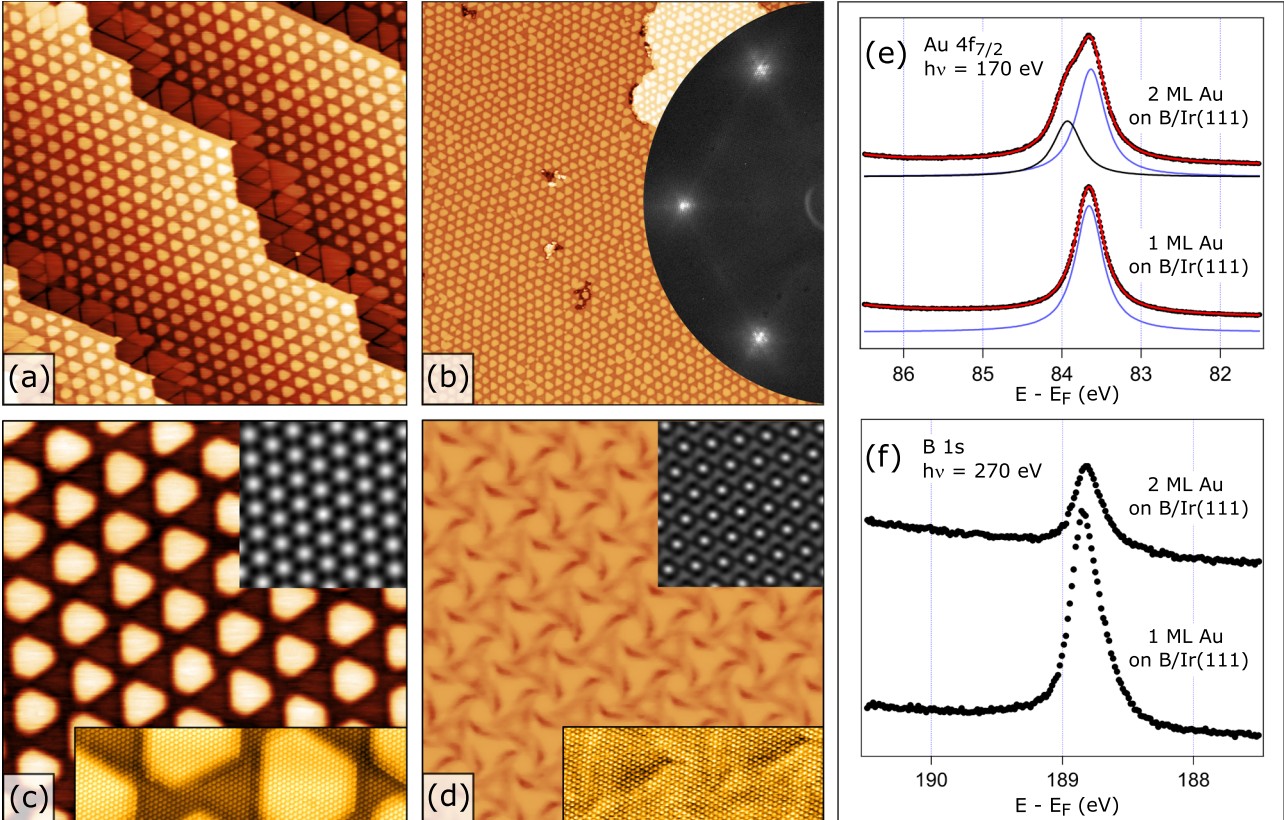

**Fig. 2 | Nano-structuring of a 2 ML thick Au film on Ir(111) by embedding interfacial B. a** 150 nm STM image of 1.9 ML Au / Ir(111) after depositing ca. 0.5 ML B at 550 °C, (**b**) 300 nm STM image of 2 ML Au / Ir(111) after depositing ca. 1 ML B at 550 °C showing long-range ordering, inset: LEED pattern from this surface taken at E = 70 eV, (**c**) same as (**b**) but 45 nm image, inset top: auto-correlation function from this image, inset bottom: 20 × 6 nm STM image with atomic resolution, (**d**) same as (**c**) but without interfacial B, inset top: auto-correlation function from this image, inset bottom: 15 × 6 nm STM image with atomic resolution. The colour scale (height scale) is made the same in (**c**) and (**d**) to emphasize a large difference in the corrugation heights. **e** Au $4f_{7/2}$ photoelectron spectra taken with the photon energy of 170 eV from single- and double-atom thick Au layers on B/Ir(111), (**f**) B 1s photoelectron spectra taken with the photon energy of 270 eV from single- and double-atom thick Au layers on B/Ir(111). In (**e**), red curve is a result of the peak fitting; black and blue curves represent peak fit components from the buried and the topmost Au layers, respectively. In (**e**–**f**), black dots represent experimentally measured data points.

consistent with what we observe in the spectrum from 2 ML Au film on B/Ir(111) in Fig. 2e.

Whether the single- or double-layer Au films on Ir(111) are exposed to the B flux at elevated temperatures and form the respective periodic patterns, the binding energy of the B 1s XP spectra remain nearly identical for those samples (refer to Fig. 2f). However, the intensity of the B 1s XP spectra decreases significantly for the 2 ML thick film, suggesting it is attenuated in a manner similar to that of the bulk-like Au atoms. It is therefore plausible to assume that in both cases B atoms penetrate through the Au films at elevated temperatures and become anchored at the Au/Ir interface upon sample cooling forming a buried B layer ultimately responsible for the formation of Au nanostructures.

**Buried B layer structure distinct from borophene**

On the pristine Ir(111) surface, boron is known to form a χ-type borophene structure at elevated temperatures[20,23]. This borophene is corrugated, with a striped appearance in each of the three rotational domains and a hexagonal hole density of 1/6 (defined as a ratio of number of single-atom vacancies in the triangular lattice to the total number of triangular lattice sites within a unit cell). It induces a (2 × 6) reconstruction on Ir(111), and its formation can be easily recognized in LEED and STM[20].

In the case of ultrathin Au films on Ir(111) exposed to the B flux upon heating, it has been established by the Ir 4ƒ XPS (see Fig. 1g) that the B atoms penetrate through the Au film and form a buried layer at the Au/Ir interface. Therefore, it is natural to assume that this buried B

layer adopts the above-mentioned χ-type borophene structure, as shown in the model presented in Fig. 3h. To demonstrate the difference in appearance of the χ-type borophene on Ir(111) and the B layer buried at the Au/Ir interface, Fig. 3a shows an STM image from the Ir surface covered with approximately 0.3 ML Au and 0.6 ML B. The image reveals two distinct areas: with characteristic stripy borophene motif and with a 1 ML thick Au film nanostructured by the interfacial B. This raises the question: Is the structure of the buried B layer identical to the χ-type borophene, identical to another borophene allotrope, or perhaps not a borophene at all?

The first clear indication comes from the comparison of the surface-sensitive B 1s XPS spectra. The spectrum obtained from a freshly grown χ-type borophene on Ir(111) can be reliably fitted using three distinct components B1, B2 and B3 (Fig. 3b), with the BE of 188.85 eV, 188.36 eV and 188.53 eV, respectively. This spectrum closely matches recent XPS data reported for this 2D boron sheet[24]. The (2×6) unit cell of this borophene contains 25 B atoms in a considerably corrugated structure[20]. Therefore, B1 to B3 actually mimic three groups of components: B1 reflects a distribution of B atoms in the most intimate contact with Ir, B2 corresponds to the uppermost B atoms gaining extra electron charge, while B3 represents a distribution of B atoms in-between these two extremes. It is worth noting that this spectrum is highly sensitive to any sample imperfections, like small amounts of O contamination (see Supplementary Fig. 4 and ref. 24) or B atoms dissolved in the Ir substrate, which may give a peak at 187.4 eV.

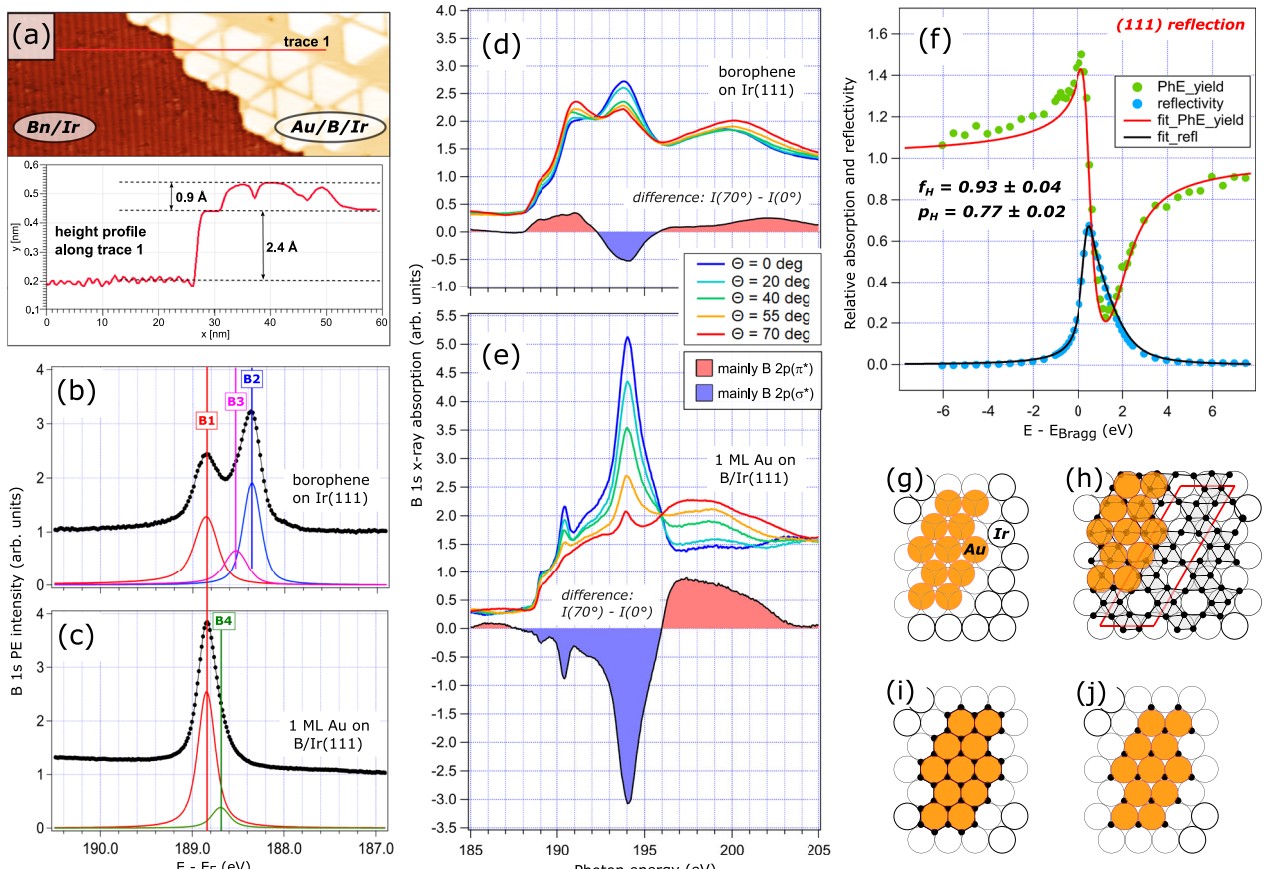

**Fig. 3 | Revealing the structure of the buried interfacial boron. a** 80 × 30 nm STM image with two co-existing structures: χ-type borophene on Ir(111) and 1 ML Au on Ir(111) nano-structured by the underlying B layer. Height profile along trace 1 is shown below. **b–c** B 1 s X-ray photoelectron spectroscopy (XPS) measured with the photon energy of 270 eV from the χ-type borophene on Ir(111) (**b**), and the B layer embedded at the interface between 1 ML Au and Ir(111) (**c**). **d–e** B K-edge near-edge X-ray absorption fine structure (NEXAFS) spectra taken as a function of angle Θ between the surface plane and the direction of photon polarization from the χ-type borophene on Ir(111) (**d**), and the B layer embedded at the interface between 1 ML Au and Ir(111) (**e**). Differences between spectra taken at Θ = 70° and 0° are shown as

colour-filled areas in both (**d**) and (**e**). **f** Measured normal-incidence X-ray standing wave (NIXSW) absorption (B 1 s photocurrent yield, green dots) and reflectivity (blue dots) profiles from 1 ML Au on B/Ir(111), along with their fits (red and black curves, respectively). **g–j** Structural models of 1 ML Au on Ir(111): (**g**) without boron interlayer, (**h**) with a χ-type borophene as an interlayer, (**i**) with a honeycomb borophene as an interlayer, (**j**) with B atoms at the hcp (alternatively fcc) positions as an interlayer. Models (**h–j**) are simplified by making the Au-Au distance equal to the Ir-Ir distance ((1 × 1) unit cell approximation). Ir, Au and B atoms are shown as white, orange and black circles, respectively. In (**h**) the (2 × 6) borophene unit cell is outlined in red.

The B 1 s XP spectrum is clearly different for the B layer buried at the Au/Ir interface (Fig. 3c). The major component, B1, remains at the same BE of 188.85 eV, while a small new component, B4, emerges at 188.69 eV, and components B2 and B3 are seemingly absent. This suggests that only the most strongly bonded B species of the borophene sheet survive as a buried interlayer in the Au/B/Ir hetero-structure. Moreover, the reduction of the full width at half maximum (FWHM) of the B1 component from 300 meV to 230 meV in going from pristine χ-type borophene (Fig. 3b) to the buried B layer (Fig. 3c) indicates a transition from a collection of energetically similar B1 boron species to only one. As the chemical nature of the buried B layer differs significantly from that of the χ-type borophene, its atomic structure should differ too. Indeed, the reduction in the number of B species on the surface would suggest a flatter B layer, an assumption confirmed unambiguously by the structural measurements below. The precise origin of components B1 and B4 will be discussed later in conjunction with the proposed structural model.

A striking structural difference between these two B films is also visible in their B K-edge near-edge X-ray absorption fine structure (NEXAFS) spectra (Fig. 3d-e). By varying the angle between the surface plane and the polarization vector of incoming radiation, the symmetry of the low-lying unoccupied B 2p states can be revealed, distinguishing

between the out-of-plane B $2p_z$(π*) and the in-plane B $2p_{x,y}$(σ*) con-tributions. The angle-dependent B K-edge NEXAFS spectra from the χ-type borophene (Fig. 3d) show unoccupied B 2p(π*) states at 188-192 eV and a broad B 2p(σ*) resonance at 194 eV. This is particularly evident from the curve representing intensity difference recorded at normal and grazing incidence, where positive and negative values represent the domination of π* and σ* states, respectively. In contrast, the spectra from the B layer buried at the Au/Ir interface (Fig. 3e) show no indication of B 2p(π*) states. Instead, only B 2p(σ*) states are pre-sent, with pronounced resonances at 190.5 eV and 194 eV. The absence of unoccupied B 2p(π*) states suggests that the B $2p_z$ orbitals are essentially filled, while the new sharp B 2p(σ*) resonances indicate a significantly different in-plane local environment for the absorbing B atoms. Furthermore, the overall angle dependence is much stronger in Fig. 3e than in Fig. 3d (note the same intensity scale), implying that the interfacial B layer is significantly flatter compared to the corrugated χ-type borophene.

Further insights into the structure of the interfacial B layer can be obtained through analysis of normal-incidence X-ray standing waves (NIXSW) data. By fitting the B 1 s photoelectron intensity across the Bragg energy for the Ir(111) reflection (Fig. 3f), we can determine the values of the coherent fraction ($f_H$) and the coherent position ($p_H$) for

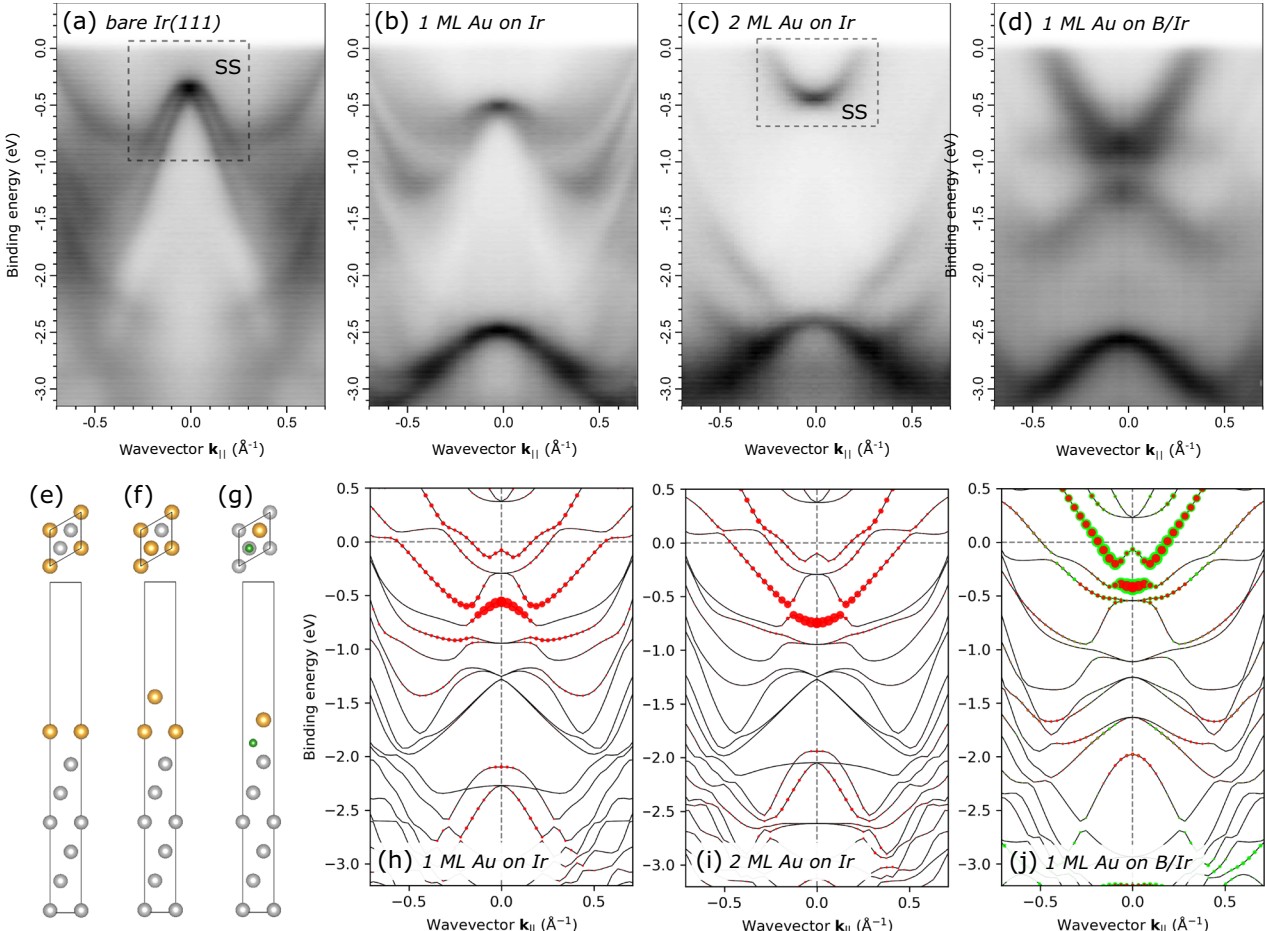

**Fig. 4 | Electronic structure of 2D metal films from angle-resolved photoelectron spectroscopy (ARPES) and Density Functional Theory (DFT). a–d** ARPES along Γ – K direction of the surface Brillouin zone (SBZ) of Ir(111) taken with the photon energy of 60 eV from (**a**) pristine Ir(111), (**b**) 1 ML Au on Ir(111), (**c**) 2 ML Au on Ir(111), (**d**) 1 ML Au on B/Ir(111). "SS" stands for "surface state". **e–g** Top and side view of the geometry-optimized slab models used in the band-structure calculations: (**e**) 1 ML Au on Ir(111), (**f**) 2 ML Au on Ir(111), (**g**) 1 ML Au on B/Ir(111). Colour coding: Ir atoms are grey, B atoms are green, Au atoms are orange. **h–j** Calculated band structures along Γ – K direction of the SBZ of Ir(111), (**h**), (**i**) and (**j**) are calculated for the models (**e**), (**f**) and (**g**), respectively. Contributions from the Au $6p_z$ states and B $2p_z$ states are highlighted with red and green circles, respectively.

the B atoms in the buried interfacial layer. A very high $f_{111}$ value of $0.93 \pm 0.04$ indicates that all the B atoms reside essentially within a single plane with respect to the surface, confirming that the interfacial B layer is indeed flat. The measured $p_{111}$ value of $0.77 \pm 0.02$ suggests an adsorption height of $1.70 \pm 0.04$ Å above the Ir(111) surface (ignoring relaxation of the topmost Ir layers), which is indicative of a strong interaction between Ir and B. In an attempt to elucidate the B atom positions on the Ir(111) surface by triangulation, NIXSW measurements were also performed for the (1$\bar{1}$1) reflection of the Ir crystal (refer to Supplementary Fig. 5). In this case, the low $f_{1\bar{1}1}$ value of $0.52 \pm 0.03$ suggests the co-existence of at least two sites for B atoms. An analysis of the $f_{1\bar{1}1}$ and $p_{1\bar{1}1}$ values (for details refer to the SI) allows us to conclude that B atoms may reside in both the hexagonal closed-packed (HCP) and the face-centred cubic (FCC) hollow sites of Ir(111), with the HCP:FCC occupation ratio close to 2:1.

Summarizing the findings from XPS, NEXAFS and NIXSW techniques, it becomes evident that B atoms at the Au/Ir interface can be characterized by an identical or very similar chemical state. They are strongly bonded to Ir, and are uniformly located at the same height, most probably in the hollow sites of the Ir(111) surface. Consequently, we can exclude the χ-type borophene (Fig. 3h) from further consideration. However, as both HCP and FCC adsorption sites seem possible, it is difficult to say at this stage whether these sites may co-exist as nearest neighbours, forming a honeycomb B interlayer (Fig. 3i),

or exist solely as either HCP or FCC individual domains (Fig. 3j). As detailed below, theoretical calculations will assist in establishing the final structural model.

## Signatures of 2D metal in the band structure

It is anticipated that the detachment and nanopatterning of the gold monolayer from the underlying metal substrate can alter its electronic properties. The angle-resolved photoemission spectroscopy (ARPES) technique has been employed to reveal differences in the band structure between the nanopatterned Au ML on the B/Ir substrate and the Au ML in direct contact with the Ir metal. In Fig. 4, a portion of the measured band structure at the Γ point of the surface Brillouin zone (SBZ) of Ir(111) in the Γ−K direction is shown for several scenarios of surface modification.

At the pristine Ir(111) surface (Fig. 4a), the structure is dominated by a characteristic Rashba-type spin-orbit split surface state (SS) dispersing downwards from the Γ point, where it is primarily composed of Ir $6p_z$ orbitals from the topmost Ir atoms, consistent with the previously reported data[25]. Once a single Au ML is formed atop Ir(111), this SS is effectively suppressed, and a distinct Au $5d$ band forms located at 2.5 eV at the Γ point and dispersing downwards (Fig. 4b). With a 2 ML thick Au film, a new SS starts to emerge at 0.5 eV at the Γ point, exhibiting an upward dispersion (Fig. 4c). This appears to be a precursor of the well-known Au(111) Shockley SS with the Au $6p_z$ orbital character[26].

Notably, no characteristic Rashba-type spin-orbit splitting of the Au(111) SS can be observed for the 2 ML thick Au film, probably due to the Au layer thickness being below the SS penetration depth (~3.2 monolayer, see ref. 27) and insufficient for complete formation of the SS. As can be seen in Fig. 4d, the most intriguing transformation occurs for the Au ML atop B/Ir(111), where a strongly upward-dispersing band appears at 1.3 eV at the Γ point, in great contrast to the similar system lacking a B interlayer (b).

The origin of this band could be understood by comparing ARPES data with the theoretically calculated band structure derived from an accurate atomic model. However, the precise model is not known a priori. Moreover, even if it were known, it would contain a huge number of atoms, rendering quantum mechanical calculations too challenging. Indeed, the surface supercell in the 1 ML Au/B/Ir structure (Fig. 1d) has a period of 11.7 ± 0.1 nm, equivalent to 43 Ir-Ir interatomic distances, resulting in 1849 Ir atoms in each layer. With several Ir layers, one monolayer of B, and one ML of Au, the total number of atoms in the slab would surpass $10^4$, thus making exact band structure calculations beyond the reach as of today.

Therefore, to facilitate relevant band-structure calculations, it is necessary to approximate the actual models with much smaller slabs. The simplest approach is the (1 x 1) unit cell approximation, which assumes an identical (or multiple thereof) number of atoms for each element per atomic layer. This approach disregards any moiré structures, and may create unrelieved strain in the adsorbate films, leading potentially to shifts and distortions of individual bands. Despite these limitations, it is anticipated that these (1x1) slab calculations will offer qualitative insight into the band characteristics.

The (1x1) unit cell slabs underwent geometry optimization prior to band-structure calculations. Additionally, they were examined for convergence concerning the number of Ir layers (up to 12 Ir layers) and the vacuum spacing within the slabs (up to 30 Å of the vacuum space). The resulting models are shown in Fig. 4 (e-g), while the atomic coordinates can be found in the SI.

As previously demonstrated by STM and LEED (refer to Supplementary Figs. 1 and 2), a single Au ML on Ir(111) grows pseudomorphically. Therefore, the (1x1) model in Fig. 4e is expected to yield a band structure, which is not only qualitatively correct but also quantitatively accurate. Indeed, the agreement between the experimental (Fig. 4b) and calculated (Fig. 4h) band structure is very good in this case. In particular, the contribution of the Au $6p_z$ states (highlighted by red circles in Fig. 4h) is seen to be responsible for the observed modification of the Ir(111) surface state by the Au ML in Fig. 4b. Similarly, in the case of 2 ML Au on Ir(111), the (1x1) unit cell calculation (Fig. 4i) can reasonably reproduce the experiment (Fig. 4c), despite the fact that this film should already be somewhat strained (its relaxation results in a moiré structure visible in Fig. 2d). The parabolic upward dispersion of the Au $6p_z$ states in Fig. 4i clearly mirrors the onset of the Au(111) surface state visible in Fig. 4c. Note that the majority of bands shown in Fig. 4i stem from the Ir substrate; they are no longer visible in experiment (Fig. 4c) because of the 2 ML thick Au film on top.

While the cases of 1 or 2 Au ML on Ir(111) are relatively straightforward, the incorporation of B below a single Au ML on Ir(111) presents additional complexity, as either 1 or 2 interfacial B atoms can be accommodated per (1x1) unit cell. For a single B atom, the most energetically favourable position was identified as the HCP hollow site on Ir(111), with Au atoms located above the Ir atoms of the topmost Ir(111) layer; this is the case shown in Fig. 4g. On the other hand, the FCC hollow sites for B atoms cannot be definitely excluded, because the energy difference between these two configurations is just 0.06 eV according to our DFT calculations (the relative energies of all considered atomic arrangements are listed in the SI). This means that domains with exclusively HCP or FCC B atom arrangements may potentially coexist in experiments, in agreement with indications from the NIXSW measurements for the (1$\bar{1}$1) reflection (see Supplementary

Fig. 5). In the case of two B atoms per (1x1) unit cell, a honeycomb B layer would form (as illustrated in Fig. 3i), with the most energetically favourable positions identified as the atop sites and the HCP hollow sites on Ir(111). This atomic arrangement is presented in the SI, along with the calculated band structure (Supplementary Fig. 6a). The honeycomb B interlayer in the theoretically optimized geometry demonstrates strong corrugation (0.67 Å height difference between B atoms in the HCP and atop positions), incompatible with the NIXSW findings that determined a uniform height of all B atoms above the Ir surface. Additionally, placing the corrugated honeycomb B layer at the Au/Ir interface yields a band structure very different from that observed by the ARPES shown in Fig. 4d (for the direct comparison, refer to Supplementary Fig. 6). Therefore, we exclude the honeycomb B arrangement from the consideration and instead focus on the model shown in Fig. 4g. The band structure calculated for this model is shown in Fig. 4j. A prominent feature of this structure is a strongly upward-dispersing parabolic band at 0.5 eV at the Γ point, comprising states with mixed Au $6p_z$ (red circles) and B $2p_z$ (green circles) character. The overall similarity in appearance of this feature to the strong parabolic band in the ARPES from the 1 ML Au/B/Ir system (Fig. 4d) is obvious. A certain deviation in the energy position of this band between the experiment (Fig. 4d) and the calculation (Fig. 4j) can be attributed to the above-mentioned limited applicability of (1 x 1) unit cell calculations in accurately reproducing energy bands for systems with large supercells. Indeed, as will be shown below, the real supercell contains areas where B atoms are missing, leading to additional electron doping that is not accounted for in the (1x1) unit cell calculations. To align with the experimental ARPES data in Fig. 4d, the Fermi level in Fig. 4j would need to be shifted upwards by approximately 0.6 eV.

We identify the upward-dispersing parabolic band with the Au $6p_z$ – B $2p_z$ character in the experimental band structure in Fig. 4d as a signature of essentially two-dimensional gold decoupled from the metallic substrate by the B interlayer. The calculated band structure of an Au monolayer bonded to the B/Ir(111) substrate is compared in the Supplementary Information with the band structure of the same monolayer separated from it artificially by 10 Å (see Supplementary Fig. 7 and discussion in the Supplementary Note 8). Although the Au-B interaction clearly affects the Au $6p_z$ states, the overall band structure of the levitating Au monolayer is clearly recognizable in the case of bonded monolayer. This comparison indicates that the nanostructured Au monolayer, while influenced by Au-B interactions, retains key features of a two-dimensional metallic monolayer due to the separation provided by the boron interlayer.

At this point, it is worth comparing the unoccupied part of the band structure calculated for the Au/B/Ir heterostructure with the B K-edge NEXAFS spectra discussed in Fig. 3e. As illustrated in Supplementary Fig. 8a, the B $2p_{xy}$ states constitute the dominating contribution to the unoccupied B-related DOS in the range of several eV above the Fermi level. Consequently, the prevalence of B $2p(\sigma*)$ states in the near-edge region of the B K-edge X-ray absorption spectrum collected from the Au/B/Ir heterostructure underscores the robustness of our structural model for this system.

In summarizing this part, our examination of the (1 x 1) unit cell DFT results and the ARPES data suggests that within the 1 ML Au/B/Ir system B atoms occupy HCP (or FCC) hollow positions on Ir(111), while Au atoms preferentially reside in the next layer atop the surface Ir atoms. The incorporation of B atoms at the Au/Ir interface effectively decouples the Au monolayer making its electronic structure essentially two-dimensional.

### Origin and nature of nanostructuring in Au mono- and bilayers

DFT calculations are of little assistance in explaining the nature of the peculiar nanopatterning of the Au mono- and bilayer depicted in Figs. 1 and 2, due to limitations arising from the large size of the supercell. On the other hand, a comprehensive toolbox of generic

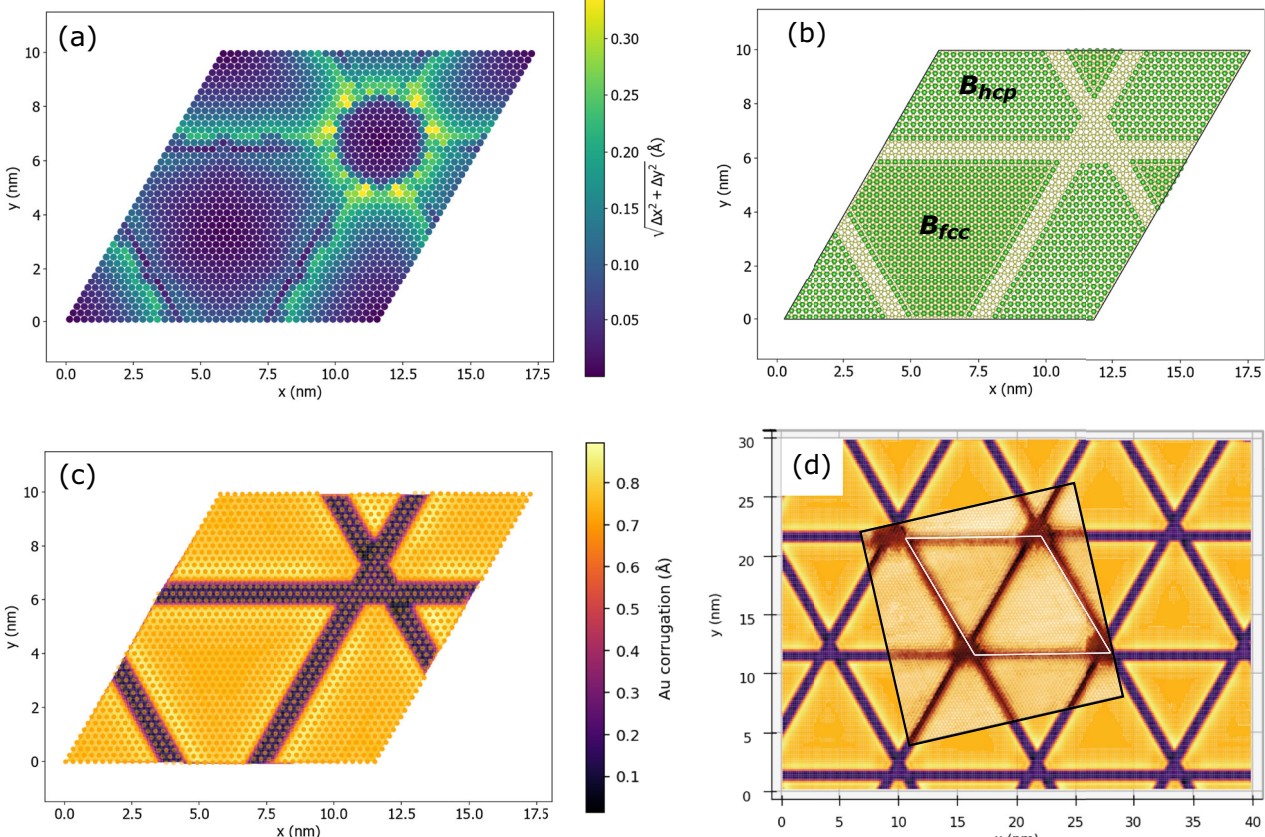

**Fig. 5 | Unravelling the nanopattern structure in monolayer gold on B/Ir(111) using generic force field calculations with periodic boundary conditions (pGFN-FF). a** Force field (FF)-optimized model with a complete B monolayer: in-plane displacement of individual B atoms from the hexagonal close-packed (HCP) hollow sites caused by the in-plane strain in the heterostructure composed of 6 layers of 43 × 43 Ir atoms, 1 layer of 43 × 43 B atoms in the HCP positions and 1 layer of 42 × 42 Au atoms. **b–d** FF-optimized model with 3 missing B rows. **b** only two

topmost Ir layers and the B layer of the Au/B/Ir heterostructure are shown; before optimization, B atoms were arranged in HCP hollow sites on the large (truncated) triangle, and in the face-centred cubic (FCC) hollow sites on the small triangle. **c** colormap showing the height corrugation in the topmost Au monolayer in the same heterostructure; positions of individual Au atoms are indicated with orange dots. **d** The Au monolayer structure from (**c**) translated over several supercells along with the superimposed STM image from Fig. 1d.

force field-based methods often approaching the accuracy of quantum-mechanical calculations has been developed for calculating atomic arrangements in large ensembles of particles, such as metal-organic frameworks and complex biomolecules with large variety of elements. In this work, we utilized a generic force-field (FF) method known as GFN-FF[28] (where GFN indicates the approach to yield reasonable Geometries, vibrational Frequencies, and Noncovalent interactions), in its specific implementation for periodic systems (pGFN-FF)[29], to simulate atomic structure within real supercells containing over $10^4$ particles. As mentioned earlier, the rhombus supercell in the case of 1 ML Au on B/Ir(111) (Fig. 1d) has a lattice parameter equivalent to 43 Ir-Ir distances. Aiming to replicate the respective moiré structure, a periodic slab for pGFN-FF geometry optimization was constructed comprising six (43 x 43) atomic layers of Ir, succeeded by a complete monolayer of B occupying the HCP hollow sites of the Ir substrate, and further followed by a (42 x 42) monolayer of Au. Given the uncertainty regarding the completeness of the buried B layer, the intention was to assess whether the strain in this heterostructure would displace some B atoms from their HCP positions, rendering them energetically unfavourable.

Figure 5a shows a monolayer of B atoms in HCP hollow sites of the substrate, sandwiched between the (43 x 43) Ir and (42 x 42) Au layers after pGFN-FF geometry optimization. The colour scale represents the level of in-plane displacement, $d = \sqrt{(\Delta x)^2 + (\Delta y)^2}$, experienced by each B atom due to in-plane stress. Apparently, some B atoms feel

considerable strain causing them to displace from the ideal HCP hollow positions. As the formation of the nanostructured Au monolayer occurs at high temperatures, it is plausible to assume that the B and the Au atoms arrange themselves into a coherent bilayer upon cooling, guided by the thermodynamics of the growth process. In this scenario, the displaced B atoms may face energetic disadvantages and hence avoid these positions altogether.

We suggest that three rows of the least energetically favourable B atoms need to be removed along each of the three main crystallographic directions to accurately reproduce trenches in STM images. However, the characteristic ring of displaced B atoms around the crossing point of these trenches (Fig. 5a) was not observed as a circular trench in experiments. We speculate that kinetically driven ripening, where larger structures grow at the expense of smaller ones, drives B atoms away from the trench-crossing points, resulting in round areas free of B atoms at the supercell corners, as seen in Fig. 1d. The second assumption regarding the B interlayer structure is based on NIXSW and STM observations. NIXSW data reveals that B atoms are located in both HCP and FCC hollow sites, with the FCC fraction close to 1/3. The distinct appearances of Au atoms in the STM images of the supercell, differing between the large and small triangles (see Fig. 1d), likely result from variations in the placement of underlying B atoms: HCP sites for the large triangle and FCC sites for the small triangle. Based on these assumptions, the following supercell slab was constructed for the pGFN-FF geometry optimization: six 43 x 43 Ir atom layers followed by an incomplete layer of B atoms and then a monolayer of Au atoms. The

optimization results of this heterostructure are shown in Fig. 5b–d, and its structure file (called "final_optimized_structure.zip") can be found in the Supplementary Materials. Figure 5b shows only the incomplete B layer and the two topmost Ir layers, highlighting the concept of the missing B rows and the two different domains of B atoms, $B_{hcp}$ and $B_{fcc}$. Here, the fraction of the $B_{fcc}$ atoms is 39%, close to the NIXSW-predicted value of 1/3. The structural model with an incomplete B monolayer suggests that the topmost Au monolayer would sink into the trenches left by the missing B atoms, contacting the Ir substrate directly. Indeed, this behaviour is reproduced in the calculated height variation of the topmost Au monolayer shown in Fig. 5c. The calculated corrugation amplitude is around 0.8 Å, in perfect agreement with both experimental STM profiles across the supercell (0.9 ± 0.1 Å, see Supplementary Fig. 3d) and the DFT-calculated height difference of the Au ML with and without a B interlayer (0.85 Å, see Supplementary Fig. 3e). The assumption that the Au ML is in contact with Ir at the trenches where B rows are missing is further supported by the survival of the Ir3 component in the Ir $4f$ XPS spectrum despite the formation of the buried B layer (Fig. 1g), implying that certain regions of the Au ML maintain direct contact with Ir(111). Overall, our model aligns well with the STM data (see the superimposed STM image in Fig. 5d), elucidating how the rhombus supercell is divided into a larger and a smaller triangle and explaining why the corners of the triangular Au nanostructures may appear not ideally sharp.

Furthermore, the structural model with three missing B rows in the (43x43) Ir supercell is consistent with the B $1s$ XPS spectra in Fig. 3c. Indeed, the B atoms on the periphery and in the middle of triangles may have slightly different chemical surrounding, resulting in the components B4 and B1, respectively. The model with 3 missing B rows in each crystallographic direction (see Fig. 5b) suggests a ratio of 17% for "edge" versus "middle" B atoms (210 versus 1236 B atoms), while the intensity ratio B4:B1 is measured at 19 ± 5% (based on several experiments and fitting procedures). The quantitative match between these values suggests that the components B1 and B4 in the B $1s$ XP spectrum indeed originate from the B atoms inside and on the edge of the boron triangles, respectively.

Although the nanopatterned Au monolayer on B/Ir(111) appears to have a degree of attachment to the Ir substrate at the trenches where B atoms are absent, the vast majority of its area is detached from the support by regularly embedded B atoms, thus representing a fair approximation to the hypothetical truly two-dimensional gold monolayer.

Similarly to the case of Au monolayer on B/Ir(111), pGFN-FF calculations were applied to the case of Au bilayer. Here, the rhombus supercell parameter of 6.7 ± 0.3 nm corresponds to 24 Ir-Ir distances. Therefore, the following slab was constructed for the calculations to visualize the strain experienced by individual B atoms: six layers of 24 x 24 Ir atoms, one layer of 24 x 24 B atoms in the HCP positions on the Ir substrate and two layers of 23 x 23 Au atoms. The in-plane displacement of the B atoms caused by the strain in this slab is shown in the colormap of Supplementary Fig. 9a. Removing the considerably displaced B atoms results in patterns like the one shown in Supplementary Fig. 9b. Regardless of the displacement threshold used for B atom removal, there is always one large area per supercell where the remaining B atoms tend to agglomerate, reproducing the experimentally observed "dotty" structure reasonably well. Note that the Au bilayer cannot be considered even nearly freestanding, due to the large area of direct contact between Au and Ir atoms.

Finally, we would like to summarize our current understanding of the factors leading to the formation of nano-patterned Au mono- and bilayers upon embedding boron at the Au/Ir interface, a mechanism that may be relevant to other metal films. Ultrathin metal layers may become strongly strained on mismatched substrates (like Au on Ir), particularly at low temperatures (like room temperature). When B atoms arrive at such an interface at elevated temperatures, the system cools and gradually freezes introducing a strain field. This field dictates where B atoms can reside and where not, elevating those areas of the metal film where boron is present. For thicker metal films (>2 ML in the case of Au on Ir) the strain at the interface is considerably relieved, hampering the formation of strictly periodic nanostructures. As the strain field is specific to each adsorbate/substrate combination, we speculate that the nanopatterning motif can vary significantly for metals other than Au on the same B/Ir substrate.

## Outlook and summary

The proposed bottom-up approach for the formation of nearly freestanding nanostructured Au monolayers not only advances our comprehension of the fundamental properties of 2D metals but also provides a platform for more practical studies. For example, the nanostructured Au films may facilitate the ordered arrangement of large molecules or various size-selected clusters at a macroscopic scale for further investigations of their catalytic, optical, or magnetic properties. Additionally, the catalytic activity of the nearly freestanding Au 2D layers themselves may differ from that in other forms of Au-based catalysts, presenting an interesting field to explore. Since the samples were prepared at relatively high temperatures, they are thermally stable up to 500°C in vacuum. With gold comprising the top layer, they can also withstand exposing to ambient conditions without a substantial chemical degradation. We brought freshly grown samples with 1 ML Au on B/Ir(111) to air and then returned them back into vacuum, still observing the characteristic triangular nanopatterns in STM, as illustrated in Supplementary Fig. 10. Hence, these samples are easy to grow due to the bottom-up growth procedure and are resilient under various temperature and vacuum conditions, rendering them a suitable substrate for a diverse range of surface-science experiments.

On a broader scale, the bottom-up approach based on the embedment of boron interlayer holds promise for the creation of nearly freestanding 2D metal films from a range of various elemental metals, such as Ag and Cu. Additionally, the nanopatterning motifs can be tailored by employing substrates with symmetries different from hexagonal. These routes can enable engineering 2D metal materials with customized functionalities and precise control over their properties.

In summary, we have devised a bottom-up strategy to produce large-scale, nearly freestanding nanostructured 2D Au monolayers by orderly embedding B atoms at the Au/Ir interface. A decoupling of the Au monolayer from the metal substrate triggers a transition in its electronic properties, reflecting a shift from 3D to essentially 2D metal bonding, as shown by ARPES and DFT. For Au bilayers, the same approach produces an alternative modulation pattern of interlinked periodic nanodots. Combining STM, synchrotron-based X-ray spectroscopies, and theoretical calculations, we unveil the structure of the buried B species and their crucial role in forming the nanostructured Au mono- and bilayers. Interfacial B atoms do not form the χ-type borophene typically observed on the Ir(111) surface but adsorb in the Ir substrate hollow sites. Strain in the Au/B/Ir heterostructures enforces the B interlayer to become incomplete, with different patterns of missing B atoms for 1- and 2-ML Au films dictating different nanostructuring motifs. The resulting Au films exhibit remarkable thermal stability, withstanding temperatures up to 500°C in vacuum and retaining resilience under ambient conditions. Their successful fabrication offers exciting prospects for advancing our understanding of 2D metallic materials. Furthermore, they may serve as stable platforms for self-assembling various macromolecules and nanoparticles, offering valuable insights into their catalytic, optical and magnetic properties.

## Methods

### Experimental

All samples were grown under ultra-high vacuum (UHV) conditions (base pressure below $2 \times 10^{-10}$ mbar) in the MAX IV STM setup featuring a variable-temperature STM (VT XA SPM from Scienta-Omicron) in the analysis chamber and a LEED optics (OCI Vacuum Microengineering Inc.) in the preparation chamber. The Ir(111) crystal surface (from SPL) was cleaned by multiple cycles of sputtering with 1 keV $Ar^+$ ions and annealing at 1000 °C until an adsorbate-free atomically clean surface with large terraces and low defect density was observed in STM, accompanied by a (1x1) hexagonal LEED pattern with sharp spots and dark background. The sample temperature was monitored using a transferable thermocouple spot-welded directly to the side of the Ir crystal and double-checked with a pyrometer for high accuracy. Gold (Au) was evaporated at room temperature from a heated crucible, and the evaporation rate was calibrated using a quartz thickness monitor. The Au films were post-annealed at 400°C to flatten the layers, because as-grown Au films on Ir(111) are known to grow in a multilayer regime[30]. It was crucial for this study to deposit either exactly 1-ML or exactly 2-ML Au film without any intermediate thickness, because samples were analysed not only microscopically but also with X-ray spectroscopies. Any phase mixing would have been visible in the spectra, contributing to uncertainty in data interpretation. Fortunately, the first and second ML of Au exhibit different morphology on Ir(111) (see Supplementary Information), making it relatively easy to determine by STM if the correct amount of gold had been deposited, and to adjust the deposition time accordingly. After pre-characterization by STM and LEED, the samples were exposed to a calibrated flux of elemental boron from a boron rod (99.999% pure, from ESPI Metals) mounted in an electron-beam evaporator (EFM-3, Focus GmbH) at a substrate temperature around 550 °C. In principle, similar results can be obtained over a wide T range, from 400 to 600 °C; higher T may result in gradual evaporation of Au from the surface, while lower T would lead to reduced interfacial ordering. Finally, the samples were cooled at a slow rate (approximately 10 °C/min) to remain closer to thermodynamic equilibrium and minimize kinetically driven imperfections in the final nanostructures.

The STM and spectroscopy studies were performed at room temperature. Bias voltages for the STM were defined as the tip bias with respect to the grounded sample. STM image processing was carried out using Gwyddion software[31]. After the STM and LEED characterization, all samples of interest were transferred for further studies with X-ray spectroscopies to a respective beamline in a vacuum suitcase, maintaining a base pressure below $5 \times 10^{-10}$ mbar. Only samples with well-defined coverage and macroscopically uniform morphology were selected for spectroscopic studies. No considerable C or O contaminations were observed by XPS after such transfer to either utilised beamline.

XPS, NEXAFS and ARPES studies were performed at the Surface and Material Science branch of the FlexPES beamline[32] at the MAX IV Laboratory in Lund, Sweden. The XPS and ARPES data were collected using a DA30-L hemispherical electron energy analyser (Scienta-Omicron), while a home-built partial electron yield microchannel plate detector (with the retardation set to $-140$ V for the B K absorption edge) was used for NEXAFS spectroscopy. A LEED spectrometer similar to the one in the STM setup was used in the spectroscopy end station to verify sample quality after the UHV transfer. The overall energy resolution in XPS was set to 12 meV (Ir $4f$, photon energy 150 eV), 15 meV (Au $4f$, photon energy 170 eV) and 45 meV (B $1s$, photon energy 270 eV), in the B K-edge NEXAFS spectra it was set to 20 meV. For ARPES, the total energy resolution was set to 40 meV (photon energy 60 eV), while the angular resolution was better than 0.2 degrees.

NIXSW measurements were performed at the I09 beamline[33] at the Diamond Light Source in Oxfordshire, UK. The NIXSW data were acquired by measuring the B $1s$ XPS intensity modulation across an 11 eV energy range centred on the energy that correspond to the (111) Bragg reflection plane of Ir substrate ($E_{Bragg} \approx 2800$ eV). Measurements were performed over multiple spots on the sample. A reflectivity curve was measured prior to each NIXSW measurement, to check the quality of each surface area and recentre the photon energy range to the measured Bragg energy for that area. The individual XP spectra of NIXSW were fitted with a numerical convolution of a Gaussian and Doniach-Sunjic[34] line shape. The non-dipolar effects in the NIXSW measurement[35] were addressed using a so-called "backwards-forwards Q-parameter"[36], which was calculated theoretically[37] using the angle between photon polarisation and the median photoelectron intensity emission angle ($\theta = 18°$).

### Theoretical

DFT calculations were performed using the Perdew–Burke–Ernzerhof (PBE) exchange-correlation functional[38] and the projector-augmented wave (PAW) method as implemented in the Vienna ab-initio simulation package (VASP)[39,40] if not stated otherwise. Dispersion effects have been considered using the D3 Grimme's parametrization[41,42]. A plane wave basis set with an energy cutoff of 500 eV was utilized. The face-centred cubic (fcc) lattice of Ir was optimized using a $18 \times 18 \times 18$ k-point mesh centred at the $\Gamma$ point for Brillouin zone sampling. The calculated Ir lattice parameter, $a = 3.8424$ Å, is in excellent agreement with its experimental value of 3.8394 Å[43]. The optimized lattice of bulk Ir was used to construct a six-layer $1 \times 1$ slab of Ir(111) surface. The periodically replicated slabs were separated by a vacuum region of ~ 15 Å. The structures of boron and gold layers on the Ir(111) surface were obtained when all B and Au atoms as well as Ir atoms in the top two layers of Ir(111) were fully relaxed until forces were less than 0.01 eV Å$^{-1}$, while the Ir atoms in the bottom 4 layers of the slab were fixed. The six-layer 43x43 slab of Ir(111) (11094 Ir atoms) with the lattice parameter of $a = 116.8317$ Å was constructed using the DFT-optimized lattice of Ir. In the case of this large supercell the structural optimization of the B and Au layers including relaxations in the top two layers of the Ir surface was performed using the generic force field for periodic boundary conditions (pGFN-FF)[29] as implemented in the General Utility Lattice Program (GULP) package v. 6.1[44].

## Data availability

Relevant data supporting the key findings of this study are available within the article and the Supplementary Information file. All raw data generated during the current study are available from the corresponding authors upon request.

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

## Acknowledgements

We acknowledge MAX IV Laboratory for time on Beamline FlexPES under proposal 20200173 and MAX IV STM setup under proposal 20220019. Research conducted at MAX IV, a Swedish national user facility, is supported by the Swedish Research council under contract 2018-07152, the Swedish Governmental Agency for Innovation Systems under contract 2018-04969, and Formas under contract 2019-02496. We acknowledge Diamond Light Source for time via the in-house beam time allotment of Beamline I09. This work was partly supported by the MEXT program: Data Creation and Utilization- Type Material Research and Development Project Grant Number JPMXP1122712807; the MEXT program for promoting researches on the supercomputer Fugaku: Data-Driven Research Methods Development and Materials Innovation Led by Computational Materials Science, JPMXP1020230327; and the Institute for Chemical Reaction Design and Discovery (ICReDD), established by the World Premier International Research Center Initiative (WPI). Calculations were performed using computational resources of the supercomputer Fugaku provided by the RIKEN Center for Computational Science (Project ID: hp230212); Institute for Solid State Physics, the University of Tokyo, Japan; and the Research Center for Computational Science, Okazaki, Japan (Project: 23-IMS-C016).

## Author contributions

The project was conceived by A.P., and the experiments were designed by A.P., N.V., and D.A.D. Samples were prepared by A.P. and N.V. STM data were acquired, analysed, and interpreted by A.P. and N.V. XPS, NEXAFS, and ARPES data were acquired by N.V. and A.P., and analysed and interpreted by A.P. NIXSW data were acquired by D.A.D. and T.L.L., and analysed and interpreted by D.A.D. under the supervision of T.L.L. DFT calculations were designed and performed by A.L. under the supervision of T.T. pGFN-FF calculations were performed by A.L. and

M.T. under the supervision of T.T. The paper was primarily drafted by A.P., with significant support from N.V., D.A.D., and A.L. All authors contributed to revising the paper.

## Funding

## Competing interests

The authors declare no competing interests.
