## [Transparent Peer Review file · Nature Communications]

Boron-Induced Transformation of Ultrathin Au Films into Two-Dimensional Metallic Nanostructures

Corresponding Author: Dr Alexei Preobrajenski

Version 0:

Reviewer comments:

Reviewer #1

(Remarks to the Author)

The authors elegantly demonstrate the levitation of Au layers by the infiltration of borophene into the Au/Ir interface. Their bottom-up route results in unique triangular nanopatterning of the Au layers. The surface and subsurface structures (topmost Au layer and borophene beneath the Au) and their electronic properties are thoroughly studied using state-of-the-art equipment and theoretical simulations. This system is promising as a platform to study various chemical and physical phenomena on the 2D Au surface. After improving by addressing the following minor concerns, I would recommend this manuscript to be published in Nature Communication.

- 1) The experimental setup and conditions for preparing Ir(111) surface and depositing 1 and 2 ML Au on Ir(111) must be described in more detail in the Methods section of the main text or supplementary information (SI). The current manuscript and SI contain no detailed information regarding the Ir preparation and the precise Au deposition with 1 and 2 ML. Moreover, if such precise deposition of Au by your setup has already been reported in previous reports, they can be cited.
- 2) For Ir 4f XP spectra, the surface (Ir2) and interface component (Ir3) are discussed. Ir2 is located at lower binding energy with respect to the bulk Ir component (Ir1), while Ir3 is slightly shifted toward a higher binding energy from Ir2. What is the physical mechanism behind the shift of Ir2 and Ir3 compared to Ir1? Does the interface component mean that the charge transfer from Ir to Au as the electronegativity of Au is higher than that of Ir? It is recommended that the authors deepen the explanation relevant to the Ir2 and Ir3 components.
- 3) In addition, the authors mention the electron transfer from Ir to B concerning the Ir2 component in XPS. The electronegativity of Ir is higher than that of B; thus, it would be reasonable to assume the charge transfer from B to Ir (although the electronegativity model might be too rough here). Clarifying this aspect would deepen the understanding of the XPS spectra.
- 4) Au4f7/2 emissions of XPS for 1 and 2 ML Au on B/Ir(111) show a characteristic peak at 83.64 eV that would originate from the surface Au monolayer. Why should the peak from the surface monolayer be at lower binding energy than the bulk Au peak?
- 5) It is evident from the combination of XPS, NEXAFS, and NIXSW that the borophene beneath the Au layer would be flat. It would be a great addition if the authors could provide a cross-sectional (S)TEM image of the Au/B/Ir sample to corroborate the authors' hypothesis of the flatter borophene (as well as confirm the one-atom-thickness of the top Au layer).
- 6) Can you easily remove the top monolayer Au from the B/Ir surface by figure-touching, scraping, or scotch-taping? Assuming the bonding of B-Ir is strong and B-Au is negligible, is it possible to transfer the Au monolayer?
- 7) Although the Au monolayer on the top of B/Ir would be electronically decoupled from the Ir metallic surface by the B interlayer, there still exists the Au-B bonding. Thus, it is counterintuitive if the band structure of the Au monolayer in the Au/B/Ir closely resembles that of the freestanding Au monolayer. The author claims the strong similarity in Figure S8. The relative structures of Au 5dxz and 6pz for both cases present some similarities, but their energy positions are distinctively different. Therefore, it is questionable if one can assert that the Au in Au/B/Ir is nearly freestanding. I suggest that the authors clarify this context further.
- 8) Concerning the calculation of the freestanding Au ML in Fig. S8, why did you use 2.899 Å for Au-Au interatomic distance? As the authors mention in the main manuscript, the 3D to 2D transition in Au would result in the shrinking of the lattice constant from 2.88 for the bulk Au to 2.76 Å for 2D Au.

Reviewer #2

(Remarks to the Author)

The authors proposed a new approach for synthesizing nearly freestanding atomically thin gold via intercalating boron atoms between gold film and iridium substrate. It is very interesting to observe the morphology evolution of gold nanostructure with boron intercalation. Additionally, the structures of the mono-/bi-layer gold and boron underneath gold nanostructures are well investigated via various methods. The topic is very attractive, but the following points should be addressed before it can be considered being published in Nature communications.

1. Based on STM images in Figure 1 and 2, the height differences between gold nanostructures and separating dark corners are $0.9 \pm 0.1 \text{ \AA}$ for monolayer Au and $0.95 \pm 0.15 \text{ \AA}$ for bilayer Au. This indicates that the intercalated boron atoms are self-confined to monolayer during intercalation. Could the authors explain why?
2. Are the interatomic distances between Au atoms same in Fig 2(c) and (d)? I am assuming that the intercalation of boron atoms in the interface will introduce more internal stress in the Au film, which will affect the interatomic distances.
3. Different regions in Fig. 3(a) should be correctly labeled. The image is taken on the same substrate terrace. But there are Au/B/Ir regions with different heights. Please explain why? This is not consistent with the point in the manuscript that only monolayer boron is underneath gold nanostructures.
4. Did the authors try to prepare freestanding trilayer gold via this method? It would be interesting to check the nanostructure evolution of gold with thicker gold layers.
5. The origin of nanostructure gold should be related to thermodynamics in the process. It would be better to check how these structures will change with different cooling rates.

Minors

1. Page 15: "Based on these assumptions, the following supercell slab was constructed for the pGFN-FF geometry optimization: six 43×43 Ir atom layers followed by an incomplete layer of B atoms and then a monolayer of A atoms." "A" should be "Au".
2. Page 5: "The NIXSW data were acquired by measuring the B1s XPS intensity modulation across an 11-eV energy range centred on the energy that correspond to the (111) Bragg reflection plane of Ir substrate ($E_{\text{Bragg}} \approx 2800 \text{ eV}$)." "11-eV" should be "11 eV".

Reviewer #3

(Remarks to the Author)

Due to the novel physicochemical properties, significant efforts have been devoted to synthesizing 2D nanomaterials. Typically, metals are not considered suitable for forming 2D materials due to the inherently isotropic nature of metallic bonding, which leads to the formation of isotropic structures. In this work, Alexei Preobrajenski et al. successfully synthesized a macroscopically large, single-atom-thick, and freestanding 2D Au monolayer. STM, LEED, XPS, NEXAFS, ARPES, and NIXSW measurements were conducted to analyze this novel structure. A lot of work has been done and the discussions are relatively comprehensive. However, some concerns should be addressed before this manuscript can be published.

1. The author claims that a monolayer ("single atomic layer" in SI) of Au will form on the Ir(111) substrate. As shown in Figure S1, the Au atoms are arranged in separated rows (or rather, lines) on the surface of Ir(111). How to understand the word "layer", maybe it is not accurate.
2. In the second line of the second paragraph on page 7. "see SI and also Figure 2d", please make clear which image in the SI is being referred to.
3. The large view of 1 ML Au formed on the Ir(111) is provided in Figure S1. I suggest that the author provide a large view of the 2 ML Au formed on the Ir(111), so that the configuration of the 2 ML Au can be concretized.
4. Is the B-induced ML Au formed uniformly or exists in some areas? For example, around the step edges and the middle of the terrace. I suggest that the author provide large-view images of the 1 and 2 ML Au formed on the B/Ir surface.
5. The author claims that "XPS can independently confirm that it is indeed one and two Au monolayers" as shown in Figure 2e. The Au monolayer on B/Ir(111) only shows one peak with a binding energy of 83.64 eV. When fully coordinated Au formed (2 ML Au), the second peak appears at 83.94 eV. As a result, when only the peak appears at 83.64 eV, we know it is indeed one monolayer. However, using the appearance of the second peak to distinguish the formation of indeed 2 ML Au is not appropriate, it is hard to say that the formed Au species is indeed 2 ML Au, 1 ML+2 ML, or 1 ML + multi-layer. Because according to the point of view in the manuscript, these situations include both fully coordinated Au atoms and surface Au atoms. So, well-prepared 2 ML Au on B/Ir will show two peaks when performing XPS testing. Inversely, using the appearance of the second peak to independently confirm that it is indeed two ML Au is insufficient.
6. This work holds the view that the B atoms dive into the Au films and form the B layer between Au and Ir responsible for the formation of Au ML. Does "dive" mean that the B atoms disperse into the lattice of Au and pass through the Au films to reach the Ir(111) surface? During the synthesis process, the Au was evaporated from a calibrated home-built source at room temperature in this work. And then, treated with a flux of elemental boron at a substrate temperature around 550 °C. Many works have reported that Au atoms undergo reconstruction with the experimental condition changes (Chem. Rev. 2012, 112, 2987–3054). So, why did the author exclude the pathway that the B deposit near the Au species on the surface of Ir(111) first, and then, the Au species reconstructed and migrated on the B layer?
7. For the line profiles in Figure S3d obtained from Figure S3a-c. Are the $Z_{0.0} \text{ \AA}$ in Figure S3a-c at the same height and located at Ir(111) surface? And the heights of 1 ML Au on B/Ir and 2 ML Au on B/Ir are both around 0.9 Å?
8. For the sentence "The intensity of the bulk-like.....mean free path (around 5 Å)" at the end of the second paragraph on page 8. Relevant references should be cited. In addition, regarding the discussions of XPS data in the manuscript, it is appropriate to provide references.
9. At the end of paragraph 4 on page 9, "(see Figure S3 in the SI and ref)". Perhaps it should be revised as "(see Figure S4 in

the SI and ref)"

10. This work used the angle-dependent B K-edge NEXAFS spectra to determine the structural difference between χ -type borophene and the B layer buried at the Au/Ir interface.

(a) The χ -type borophene show unoccupied B $2p(\pi^*)$ states at 188-192 eV. In contrast, the B $2p(\sigma^*)$ of the B layer buried at the Au/Ir interface shows pronounced resonances at 190.5 eV and 194 eV. Is there exist overlap in photon energy for B $2p(\pi^*)$ and B $2p(\sigma^*)$?

(b) The intensity of B $2p(\pi^*)$ spectra of the two kinds of B sample both increase at 188-192 eV regardless of the changes in test angle. So, does this mean that both cases exist the unoccupied out-of-plane B $2p_z(\pi^*)$.

11. The third row from the bottom on page 15. ".....monolayer of A atoms." Does the "A atoms" mean Au atoms?

12. It is indeed a very interesting phenomenon that the pseudomorphic Au species will form a regular pattern of densely packed equilateral triangles after element B treatment. And the author does a lot of work to demonstrate the existence of the B layer between Au ML and Ir(111) and the truly Au 2D structure. And I believe in these conclusions. In addition, I am curious about why this happened. A scientific issue is why element B drives Au species to evolve into a new atomic layered pattern, and what is the role B plays in the process. If the authors can give more discussions or data to settle this issue, I believe this will bring more inspiration for the synthesis of 2D metallic materials.

13. For the reference part. According to the rules of Nature Communications, the format should be checked and revised carefully.

(a) For book citations. The publisher and city of publication are required.

(b) Article titles should be in Roman text, only the first word of the title should have an initial capital.

(c) Journal names are italicized and abbreviated (with full stops) according to common usage. In addition, please check the usage of the dot in the abbreviation of the journal name.

Version 1:

Reviewer comments:

Reviewer #1

(Remarks to the Author)

The authors have diligently addressed all my concerns and questions. I recommend this article for publication.

Reviewer #2

(Remarks to the Author)

All my previous comments have been thoroughly addressed in the revised manuscript. The authors have presented a well-structured and scientifically sound study that contributes meaningfully to the field of two dimensional metals. I am pleased to recommend it for acceptance.

Reviewer #3

(Remarks to the Author)

I have carefully read the responses from the author and my concerns have been settled.

We sincerely appreciate the reviewers' positive evaluation of our work, as well as their valuable and insightful comments and suggestions. We apologize for the delay in our response, which was due to the additional analyses and calculations we undertook to further strengthen certain aspects of the discussion.

Please note that the band structures presented in Figure 4, along with Figures S6, S7, and S8 in the Supplementary Information, have been recalculated and replotted in the revised version due to a systematic error identified in the original submission, where the k-space directions were defined incorrectly.

REVIEWER COMMENTS

Reviewer #1 (Remarks to the Author):

The authors elegantly demonstrate the levitation of Au layers by the infiltration of borophene into the Au/Ir interface. Their bottom-up route results in unique triangular nanopatterning of the Au layers. The surface and subsurface structures (topmost Au layer and borophene beneath the Au) and their electronic properties are thoroughly studied using state-of-the-art equipment and theoretical simulations. This system is promising as a platform to study various chemical and physical phenomena on the 2D Au surface. After improving by addressing the following minor concerns, I would recommend this manuscript to be published in Nature Communication.

1) The experimental setup and conditions for preparing Ir(111) surface and depositing 1 and 2 ML Au on Ir(111) must be described in more detail in the Methods section of the main text or supplementary information (SI). The current manuscript and SI contain no detailed information regarding the Ir preparation and the precise Au deposition with 1 and 2 ML. Moreover, if such precise deposition of Au by your setup has already been reported in previous reports, they can be cited. –

We thank the reviewer for the remark and add more details of the experimental set up and sample preparation in the Methods section.

2) For Ir 4f XP spectra, the surface (Ir2) and interface component (Ir3) are discussed. Ir2 is located at lower binding energy with respect to the bulk Ir component (Ir1), while Ir3 is slightly shifted toward a higher binding energy from Ir2. What is the physical mechanism behind the shift of Ir2 and Ir3 compared to Ir1? Does the interface component mean that the charge transfer from Ir to Au as the electronegativity of Au is higher than that of Ir? It is recommended that the authors deepen the explanation relevant to the Ir2 and Ir3 components.

On the 5d metal surfaces, the direction and modulus of surface core-level shifts (SCLS) can differ very strongly, as was shown by Mårtensson et al. (<https://doi.org/10.1103/PhysRevB.39.8181>), and as illustrated here by a chart from this paper.

Early 5d metals (Hf, Ta) demonstrate positive SCLS, while late 5d metals (Ir, Pt, Au) – negative shifts. As the chemical bonding in all these metals is dominated by 5d electrons, it is the filling of the 5d band that makes the difference. The sign change of the SCLS's is known also across the series of 4d metals

(<http://doi.org/10.1103/PhysRevB.54.8892>). The main reason for these variations is a different occupation of d states in the surface atoms: in heavier transition metals, there is an increase in surface d electrons due to electron transfer from s and p states into d states within the surface layer, while in lighter transition metals, the decrease in surface d electrons is caused by electron flow into the vacuum (<http://doi.org/10.1103/PhysRevB.54.8892>).

[REACTED]

Regarding the effect of adsorbates on the SCLS, small changes in the surface-related component's energy usually indicate physisorption with minimal charge redistribution ([https://doi.org/10.1016/0368-2048\(95\)02532-4](https://doi.org/10.1016/0368-2048(95)02532-4)). In contrast, a more significant reduction of the SCLS in the adsorption event is a manifestation of chemisorption and the formation of chemical bond ([https://doi.org/10.1016/0368-2048\(95\)02532-4](https://doi.org/10.1016/0368-2048(95)02532-4)).

In our work we observe a small change in the Ir 4f SCLS upon Au adsorption ($E(\text{Ir}3) - E(\text{Ir}2) = 0.1$ eV), but a more pronounced variation after embedding a layer of B at the interface ($E(\text{Ir}4) - E(\text{Ir}2) = 0.28$ eV). This implies that the charge transfer is rather insignificant at the Au/Ir interface, but more pronounced at the B/Ir border.

3) In addition, the authors mention the electron transfer from Ir to B concerning the Ir2 component in XPS. The electronegativity of Ir is higher than that of B; thus, it would be reasonable to assume the charge transfer from B to Ir (although the electronegativity model might be too rough here). Clarifying this aspect would deepen the understanding of the XPS spectra.

On the Pauling electronegativity scale, the values for B and Ir are 2.04 and 2.20, respectively, thus indeed suggesting a charge transfer from B to Ir. However, other electronegativity scales suggest the opposite: the Allred Rochow scale gives values of 2.01 for B and 1.55 for Ir, while the Allen scale lists 2.05 for B and 1.68 for Ir. This indicates that these concepts can hardly be trusted for this specific issue.

In general, the Ir-B bond is a typical case of the metal-nonmetal interaction, as metalloid boron tends to behave more like a nonmetal in terms of its chemical properties. Metals, particularly transition metals, tend to donate electrons when bonding with nonmetals. Therefore, it is not unusual to assume an electron transfer from Ir to B in our system, based on the observed shift of the Ir 4f core level to higher binding energy for the topmost Ir atoms. While this is not the definite proof, because final-state effects may affect the binding energy positions, it would be rather exotic for the final-state effects to completely reverse the direction of the chemical shift in XPS. Therefore, we believe that this is the electron charge transfer from Ir to B in the ground state of

the system, which is responsible for the binding energy shifts in the Ir 4f XPS. Thankful for the reviewer's comment, we have added a clarifying paragraph addressing this issue in the revised manuscript.

4) Au_{4f7/2} emissions of XPS for 1 and 2 ML Au on B/Ir(111) show a characteristic peak at 83.64 eV that would originate from the surface Au monolayer. Why should the peak from the surface monolayer be at lower binding energy than the bulk Au peak?

Here the situation is similar to the case of Ir surface discussed above. The Au 4f peak from the surface monolayer has a lower binding energy compared to the bulk Au atoms (including 2d Au layer) because it is undercoordinated. The lower coordination means there are fewer constraints on electron movement, enabling the electrons in the conduction band to rearrange and stabilize the core hole more effectively. This improved screening reduces the energy required to remove the core electron, hence leading to a lower binding energy.

5) It is evident from the combination of XPS, NEXAFS, and NIXSW that the borophene beneath the Au layer would be flat. It would be a great addition if the authors could provide a cross-sectional (S)TEM image of the Au/B/Ir sample to corroborate the authors' hypothesis of the flatter borophene (as well as confirm the one-atom-thickness of the top Au layer). –

We appreciate the Reviewer's suggestion regarding the use of cross-sectional (S)TEM to provide further confirmation of the boron layer flatness and the one-atom-thick Au layer. While we agree that (S)TEM can be an ideal method in certain cases for confirming the geometry of buried layers, we believe that, in this particular instance, using (S)TEM would be extremely challenging and potentially unnecessary.

The preparation required for (S)TEM in this case (such as using thin Ir films on an appropriate substrate rather than a bulk Ir single crystal) would be highly complex and tedious. More importantly, additional procedures like capping the sample and ion milling could introduce uncertainties, particularly when investigating a one-atom-thick buried layer.

Additionally, we would like to clarify that the buried layer in our system is not a bonded borophene structure (like the χ -type borophene on pure Ir(111) surface), which is why it is consistently referred to as a "B layer" throughout the manuscript. We propose that this B layer consists of individual boron atoms occupying favourable adsorption sites on the Ir surface. For this reason, we have relied on the careful interpretation of indirect X-ray-based techniques (XPS, NEXAFS, NIXSW) to confirm the geometry of the buried B layer, as they provide more reliable and accurate information in this context. Note also that while XPS and NEXAFS only indicate that the B layer is flat, NIXSW provides an unambiguous quantitative measure of this flatness.

6) Can you easily remove the top monolayer Au from the B/Ir surface by figure-touching, scraping, or scotch-taping? Assuming the bonding of B-Ir is strong and B-Au is negligible, is it possible to transfer the Au monolayer? –

We appreciate the insightful comments and questions of the reviewer. In our work, we were mainly focussed on the nano-structuring of Au mono- and bilayers, which is essentially driven by the strain release induced by the B interlayer at the Au/Ir interface. Therefore, this specific substrate was an integral and necessary part of the system under study so far.

On the other hand, we agree that it would be fascinating to study whether these two-dimensional Au films could survive transferring onto another (e.g. insulating) substrate and allow for further electronic, structural and magnetic measurements. However, we are also

aware of the tremendous challenges associated with exfoliating thin films of non-layered materials. While mechanical exfoliation (as suggested by the reviewer), chemical etching or electrochemical delamination may theoretically work, transferring an Au monolayer from a B/Ir substrate would be highly problematic. Simple methods like scraping, finger-touching, or Scotch-taping would likely result in damaging or tearing the delicate monolayer due to rather weak non-directional in-plane bonding between Au atoms. Potentially, more advanced techniques like polymer-supported wet transfer (maybe further combined with intercalation or selective etching with other species) would need to be explored. Given the complexity of these methods, we believe that the exfoliation experiments would require a dedicated project and fall outside the scope of this contribution.

Once again, thanks to the reviewer for the questions inspiring interesting directions for future research.

7) Although the Au monolayer on the top of B/Ir would be electronically decoupled from the Ir metallic surface by the B interlayer, there still exists the Au-B bonding. Thus, it is counterintuitive if the band structure of the Au monolayer in the Au/B/Ir closely resembles that of the freestanding Au monolayer. The author claims the strong similarity in Figure S8. The relative structures of Au 5dxz and 6pz for both cases present some similarities, but their energy positions are distinctively different. Therefore, it is questionable if one can assert that the Au in Au/B/Ir is nearly freestanding. I suggest that the authors clarify this context further. –

We thank the Reviewer for this observation and for pointing out that our argumentation may appear unclear or insufficient to the readers. While the band dispersions of the Au ML in Au/B/Ir and a freestanding Au ML are indeed similar, their exact energy positions differ by up to 2.5 eV. This discrepancy results from the fact that the Fermi levels of systems, which are not in direct electrical contact, do not align, as they do not share the same level of electron doping. In this context, we should look on the shape and relative positions of the bands, rather than on their absolute position relative to the assumed Fermi levels.

To provide a more direct comparison, we decided to calculate the band structure for the same Au/B/Ir slab, but with the Au atom positioned far away from the surface (10 Å away from its equilibrium position). This allows us to directly compare the band structure of the Au monolayer when it is bonded to B/Ir with the scenario where it is definitely freestanding (levitating above the surface). In this case, the Au-Au distance corresponds to the Ir-Ir distance (2.715 Å), which is closer to the calculated “ideal” distance of 2.74-2.76 Å for a freestanding Au monolayer, as compared to the bulk value of 2.899 Å used originally. The result of this comparison is presented here:

In the figure, the red (violet) dots highlight contributions from the Au 6p_z (Au 5d_{xz}) bands. From this comparison it becomes clear that in the bonded system (b) the overall band structure of the Au ML retains clear similarity with the case of freestanding Au ML (a), although bands in (b) become disrupted at the K and M points due to the Au-B interaction and an orbital mixing of the Au p_z states with the B p_z states. The absolute energy positions of the Au bands are shifted upwards in (a), because the Au ML is electronically decoupled from the surface and does not share the same Fermi level. The disruption of the band structure in (b), primarily in the case of the Au p_z states, does not allow us to claim that the Au ML remains strictly freestanding. However, the shape of the Au d bands is less affected by the Au-B interaction, and in general, the Au monolayer separated from Ir(111) by buried boron species exhibits characteristics of a two-dimensional metallic monolayer. The figure above (revised Figure S7) allows the reader to see quantitatively how closely the Au monolayer in our system approximates a freestanding state.

It should also be reminded that these calculations are performed for the (1x1) unit cell, not considering the full complexity of the observed supercell. Therefore, any quantitative conclusions drawn from this comparison should be made with caution.

8) Concerning the calculation of the freestanding Au ML in Fig. S8, why did you use 2.899 Å for Au-Au interatomic distance? As the authors mention in the main manuscript, the 3D to 2D transition in Au would result in the shrinking of the lattice constant from 2.88 for the bulk Au to 2.76 Å for 2D Au. –

This is correct, and we confess there was no specific justification to use the band-structure calculation of the freestanding Au ML with the bulk constant of 2.899 Å. We did perform additional calculations with the lattice constant corresponding to the Ir-Ir distance (2.715 Å), and observed that the structure is greatly the same as for 2.899 Å, but the bands are (expectedly) shifted a bit further apart due to the increased interaction. We are certain that for the lattice constant of 2.76 Å the situation will be somewhat intermediate between these two extremes.

However, as noted in our response to point (7), we have decided to use a more instructive representation in Figure S7 (previously called Figure S8) by comparing band structure calculation when Au is in contact with the B/Ir substrate and when detached. In this way we not

only demonstrate the band structure of the freestanding Au ML, but also show how the Au 6pz – B 2pz hybridization turns off upon detaching the Au ML from the substrate.

Reviewer #2 (Remarks to the Author):

The authors proposed a new approach for synthesizing nearly freestanding atomically thin gold via intercalating boron atoms between gold film and iridium substrate. It is very interesting to observe the morphology evolution of gold nanostructure with boron intercalation. Additionally, the structures of the mono-/bi-layer gold and boron underneath gold nanostructures are well investigated via various methods. The topic is very attractive, but the following points should be addressed before it can be considered being published in Nature communications.

1. Based on STM images in Figure 1 and 2, the height differences between gold nanostructures and separating dark corners are $0.9 \pm 0.1 \text{ \AA}$ for monolayer Au and $0.95 \pm 0.15 \text{ \AA}$ for bilayer Au. This indicates that the intercalated boron atoms are self-confined to monolayer during intercalation. Could the authors explain why? –

Indeed, the embedding of boron at the Au/Ir interface is a self-limiting process resulting in only one monolayer of B atoms being trapped under the Au in the bright areas of the STM images in Figures 1 and 2. In the process of cooling the sample, B atoms start forming chemical bonds with the topmost Ir layer, preferentially occupying the most energetically favourable adsorption sites (either hcp or fcc hollows in our case). Once these sites are filled completely, any excess B atoms can no longer stick to the Ir surface. As a result, they either dissolve into the Ir bulk or precipitate onto the Au surface, or both, depending on the sample cooling kinetics. Both cases have been observed experimentally. Whenever an excess of boron is added to the system, its dissolution in iridium can be seen as an extra peak in the B 1s XPS spectra at a binding energy of 187.5 eV, and its precipitation on the Au surface is visible in STM as bright protrusions populating mainly the trenches of the Au nanostructures.

2. Are the interatomic distances between Au atoms same in Fig 2(c) and (d)? I am assuming that the intercalation of boron atoms in the interface will introduce more internal stress in the Au film, which will affect the interatomic distances. –

The Au-Au interatomic distances are indeed slightly different. Although we are cautious about claiming this based on direct comparison of interatomic distances measured in different STM experiments, due to the need for extremely high scanning stability and the absence of any drift to measure these distances precisely, there is a reliable workaround based on the fact that it is much easier to measure supercell parameter values with high precision.

For a 2 ML Au film on Ir(111), the supercell parameter is $7.4 \pm 0.2 \text{ nm}$, while the insertion of a B monolayer at the interface changes it to $6.7 \pm 0.3 \text{ nm}$. Given the Ir-Ir distance of 2.72 \AA , these two supercells must be composed on each side of (closest numbers) 27 Ir / 26 Au atoms and 24 Ir / 23 Au atoms, respectively. Therefore, the Au-Au distances in the topmost (second) layer of the Au film are $2.72 \times 27 / 26 = 2.82 \text{ \AA}$ for the bilayer without buried boron, and $2.72 \times 24 / 23 = 2.84 \text{ \AA}$ for the bilayer with B. This indicates that the strain release from the pseudomorphic value of 2.72 \AA is slightly stronger with the buried B layer present.

3. Different regions in Fig. 3(a) should be correctly labelled. The image is taken on the same substrate terrace. But there are Au/B/Ir regions with different heights. Please explain why? This is not consistent with the point in the manuscript that only monolayer boron is underneath gold nanostructures. -

The main purpose of this image was to demonstrate how borophene on Ir(111) can coexist with boron buried under 1 ML of Au when the gold monolayer is incomplete. We apologize for the confusion regarding this image, as it seems really misleading without further explanation, which was omitted in the initial submission.

What was not mentioned there is that this image does not depict a single flat Ir terrace. Instead, the underlying Ir substrate has two distinct areas with a height difference of one atomic step. This can be seen in the height profile across the image, as shown in the figure above. The grey area under the profile represents the underlying Ir surface with two distinct step heights. This is not uncommon that after several sputtering and annealing cycles of the Ir(111) surface, monoatomic Ir islands can be seen on otherwise flat Ir terraces. To avoid explaining this in the main text and to prevent further confusion, we have reduced the area of this image in Figure 3 to demonstrate only two distinct areas on one and the same terrace: with borophene on Ir and with 1 ML Au on B/Ir. We also have added a line profile to convey our message more clearly.

4. Did the authors try to prepare freestanding trilayer gold via this method? It would be interesting to check the nanostructure evolution of gold with thicker gold layers. -

Yes, we tried to grow nanostructured Au films made of 3 and more monolayers of gold by embedding boron at the Au/Ir interface. However, it proves hard to enforce one specific nanostructure pattern across large-area domains in these cases. For 1 and 2 ML of Au, the strain in the films is considerable, and the intercalated boron becomes locked by this strain field forming periodic nanostructures. For thicker Au films the strain becomes partly released, resulting in a less ordered arrangement of boron at the interface. Typically, various quasi-periodic nano patterns can be observed simultaneously in STM images from such samples, as shown in the 250 nm image to the right. Notice that areas with 2 ML thick Au films (outlined by a dashed yellow line) demonstrate good periodicity, while regions with thicker Au films (3+ ML) lack strict periodicity and show nanostructures of different sizes and shapes.

5. The origin of nanostructure gold should be related to thermodynamics in the process. It would be better to check how these structures will change with different cooling rates. -

It is true that the cooling rate is a critical factor for the growth. In general, at slow cooling rate the system has better chances to reach thermodynamic equilibrium, resulting in a more ordered structure defined by the lowest total energy. The nanostructured gold films discussed in our paper were typically obtained at a cooling rate of 10 °C/min or slower, and at these rates the resulting structures were always the same, determined by the thermodynamics. Although we did not perform systematic studies of various cooling rates, we noticed that abrupt cooling resulted in a less ordered final morphology with high number of defects and irregularities. This is because in the case of fast cooling the kinetic effects start to play a role, and the system may become trapped in a non-equilibrium state. As a rule, we avoided preparations determined by the kinetics of the cooling process. In the revised version of the manuscript, we mention this fact and state the typically used cooling rate.

Minors

1. Page 15: “Based on these assumptions, the following supercell slab was constructed for the pGFN-FF geometry optimization: six 43x43 Ir atom layers followed by an incomplete layer of B atoms and then a monolayer of A atoms.” “A” should be “Au”.

2. Page 5: “The NIXSW data were acquired by measuring the B1s XPS intensity modulation across an 11-eV energy range centred on the energy that correspond to the (111) Bragg reflection plane of Ir substrate ($E_{\text{Bragg}} \approx 2800 \text{ eV}$).” “11-eV” should be “11 eV”.

We are very grateful to the Reviewer for spotting out these issues, and correct the captions as pointed out by the reviewer.

Reviewer #3 (Remarks to the Author):

Due to the novel physicochemical properties, significant efforts have been devoted to synthesizing 2D nanomaterials. Typically, metals are not considered suitable for forming 2D materials due to the inherently isotropic nature of metallic bonding, which leads to the formation of isotropic structures. In this work, Alexei Preobrajenski et al. successfully synthesized a macroscopically large, single-atom-thick, and freestanding 2D Au monolayer. STM, LEED, XPS, NEXAFS, ARPES, and NIXSW measurements were conducted to analyze this novel structure. A lot of work has been done and the discussions are relatively comprehensive. However, some concerns should be addressed before this manuscript can be published.

1. The author claims that a monolayer (“single atomic layer” in SI) of Au will form on the Ir(111) substrate. As shown in Figure S1, the Au atoms are arranged in separated rows (or rather, lines) on the surface of Ir(111). How to understand the word “layer”, maybe it is not accurate. –

We thank the reviewer for this observation and would like to clarify our use of the term “layer”. In our work, we do refer to a complete closely packed layer of gold on the Ir(111) substrate. The right panel in Fig. S1 shows an atomically resolved image of the single Au layer on Ir in the vicinity of the stripes. Also, the centre panel in Fig. S1 shows an almost complete flat Au layer, with a few black depressions, which we attribute to holes in the gold film / patches of Ir that

remain uncovered by Au. Also, the LEED patterns in Fig. S2 clearly show that the 1st Au ML on Ir(111) grows pseudomorphically, as only the (1x1) spots are visible at this Au layer thickness.

As for the bright stripes, we believe they are due to the pseudomorphic growth of the Au film on Ir(111), which induces considerable strain in it. The strain accumulated in the monolayer needs a release and may force a slight reconstruction of it. The height variation across the bright stripes is very small, below 0.15 Å, as can be seen in the revised version of Fig. S1, where we show the corresponding height profile. This indicates that although the strained Au monolayer may appear slightly “wavy”, it remains a monolayer. We apologize for not describing this clearly in the original submission; now we have added a clarification in the description of Figure S1 in the Supplementary Information.

2. In the second line of the second paragraph on page 7. “see SI and also Figure 2d”, please make clear which image in the SI is being referred to. –

We apologize for the confusion and modify the text as follows (“see Figure 2d and Figures S1b and S3c in the SI”).

3. The large view of 1 ML Au formed on the Ir(111) is provided in Figure S1. I suggest that the author provide a large view of the 2 ML Au formed on the Ir(111), so that the configuration of the 2 ML Au can be concretized. –

We thank the reviewer for this recommendation, and follow it by placing a large field-of-view image of the 2 ML thick Au film on Ir(111) into Figure S1 of the SI as a new panel, Figure S1b. We have also added an atomically resolved close-up image from the 2ML thick film within the same panel, to further facilitate a direct comparison with the 1ML thick film shown in Figure S1a.

4. Is the B-induced ML Au formed uniformly or exists in some areas? For example, around the step edges and the middle of the terrace. I suggest that the author provide large-view images of the 1 and 2 ML Au formed on the B/Ir surface. –

If the amounts of B and Au are correctly maintained, the temperature is carefully controlled, and the cooling rate is kept low, the nanostructured films grow homogeneously and uniformly across the sample surface. To illustrate this, we have already presented large-scale images (300 x 300 nm) in Figures 1b (1 ML Au on B/Ir) and 2b (2 ML Au on B/Ir). While we did not scan areas larger than this, we have a considerable collection of images at scales from 150 to 300 nm, taken from a variety of samples and from different spots on the sample surface. Based on these data, substrate step edges do not seem to represent an obstacle for the formation of uniform films. However, we acknowledge that if the atomic ratio of components is not optimal, step edges may play a considerable role. For example, by adding boron to the Au/Ir system stepwise, we have observed that it tends to intercalate first at areas close to the step edges, spreading gradually to the terrace area with increasing boron content. This situation is illustrated by the 150 nm STM image to the right, showing a predominantly 2 ML Au film on

Ir(111) with a B content close to 0.2 ML, along with a schematic representation of the underlying substrate steps. In this boron-deficient case the nanostructuring starts at the step edges, indicating that B atoms prefer step sites for the initial intercalation.

5. The author claims that “XPS can independently confirm that it is indeed one and two Au monolayers” as shown in Figure 2e. The Au monolayer on B/Ir(111) only shows one peak with a binding energy of 83.64 eV. When fully coordinated Au formed (2 ML Au), the second peak appears at 83.94 eV. As a result, when only the peak appears at 83.64 eV, we know it is indeed one monolayer. However, using the appearance of the second peak to distinguish the formation of indeed 2 ML Au is not appropriate, it is hard to say that the formed Au species is indeed 2 ML Au, 1 ML+2 ML, or 1 ML + multi-layer. Because according to the point of view in the manuscript, these situations include both fully coordinated Au atoms and surface Au atoms. So, well-prepared 2 ML Au on B/Ir will show two peaks when performing XPS testing. Inversely, using the appearance of the second peak to independently confirm that it is indeed two ML Au is insufficient. –

We partly agree with the reviewer that using XPS alone to determine the exact thickness of a multilayer can be tricky, as this method relies on a few key assumptions: layer-by-layer growth, low substrate roughness, and good knowledge of the inelastic mean free path (IMFP) of the photoelectrons. In the case of 2 ML Au films on Ir(111), we know from STM that the growth is layer-by-layer and the roughness is insignificant. The IMFP of electrons with a kinetic energy of 80 eV (photon energy was 170 eV) in bulk gold can be estimated as 3.9 Å using the method suggested in [H. Shinotsuka, S. Tanuma, C. J. Powell, D. R. Penn, Surf. Interf. Anal. 47 (2015) 871]. Given the interlayer distance of 2.35 Å for Au(111) atomic planes, the intensity of the Au 4f component from the second (lower lying) Au layer is expected to be 55% of the intensity from the top Au layer. This is consistent with what we observe in the spectrum from 2ML Au film on B/Ir(111) in Figure 2e. Therefore, XPS does suggest that the Au film is made of precisely 2 atomic layers, within the uncertainties mentioned above. In this specific case we are even more certain, because the samples were pre-characterized by STM, where rather homogeneous morphology corresponding to 2ML of Au on B/Ir(111) was observed at all tested sample areas, with the fraction of 1 ML and 3 ML coverage being well below 10%. We are thankful to the reviewer for mentioning this issue and have added the above estimate of the coverage from XPS into the revised version of the paper.

6. This work holds the view that the B atoms dive into the Au films and form the B layer between Au and Ir responsible for the formation of Au ML. Does “dive” mean that the B atoms disperse into the lattice of Au and pass through the Au films to reach the Ir(111) surface? During the synthesis process, the Au was evaporated from a calibrated home-built source at room temperature in this work. And then, treated with a flux of elemental boron at a substrate temperature around 550 °C. Many works have reported that Au atoms undergo reconstruction with the experimental condition changes (Chem. Rev. 2012, 112, 2987–3054). So, why did the author exclude the pathway that the B deposit near the Au species on the surface of Ir(111) first, and then, the Au species reconstructed and migrated on the B layer? –

We thank the reviewer for this insightful remark. Possibly “dive” is a misleading word in this context, and we have changed the corresponding sentence in the revised version. In our view, once the substrate surface is covered with a complete mono- or bilayer of Au, the B atoms must penetrate through this film to reach the Au/Ir interface. At elevated temperatures, the small B

atoms can easily dissolve in the Au lattice. In fact, in a separate experiment we observed that much more boron was needed to grow any borophene on the Au(111) surface compared to the Ir(111) surface, at the same temperature around 550 °C. This is because boron is highly soluble in Au and readily dissolves into the bulk at high temperatures.

Therefore, this pathway of B penetration through Au seems well-supported for this type of preparation. However, the reviewer has right that other pathways may also be correct. For example, we could achieve the same final structure by growing borophene on Ir(111) first, and adding correct amount of Au at 550 °C then. What ultimately matters for the growth of nanostructured Au films are the right amount of B and Au, proper annealing temperature, and slow cooling.

7. For the line profiles in Figure S3d obtained from Figure S3a-c. Are the Z₀ 0.0 Å in Figure S3a-c at the same height and located at Ir (111) surface? And the heights of 1 ML Au on B/Ir and 2 ML Au on B/Ir are both around 0.9 Å? –

In the series presented in Figure S3a-c, there is no Ir(111) surface exposed to the STM tip; all regions (both bright and dark) correspond to areas of the Au films. Therefore, the line profiles reflect the B-induced corrugation of the Au film surface in (a) and (b), and the natural corrugation of the 2 ML thick Au film on Ir(111) without boron in (c). The reference height in each profile (Z=0) corresponds to the minimum height of the respective image. The height difference of approximately 0.9 Å is caused by the presence (bright areas) or absence (dark areas) of boron under the Au films. This value is the same for the 1 ML and 2 ML thick Au films.

8. For the sentence “The intensity of the bulk-like.....mean free path (around 5 Å)” at the end of the second paragraph on page 8. Relevant references should be cited. In addition, regarding the discussions of XPS data in the manuscript, it is appropriate to provide references.

This sentence is replaced now with a new paragraph, as explained in the reply to question (5), including a new reference. Also, in the discussion of XPS chemical shifts several references to pioneering contributions in the field have been added.

9. At the end of paragraph 4 on page 9, “(see Figure S3 in the SI and ref)”. Perhaps it should be revised as “(see Figure S4 in the SI and ref)” –

We thank the reviewer for noticing this error and modify the text accordingly.

10. This work used the angle-dependent B K-edge NEXAFS spectra to determine the structural difference between χ -type borophene and the B layer buried at the Au/Ir interface.

(a) The χ -type borophene show unoccupied B 2p(π^*) states at 188-192 eV. In contrast, the B 2p(σ^*) of the B layer buried at the Au/Ir interface shows pronounced resonances at 190.5 eV and 194 eV. Is there exist overlap in photon energy for B 2p(π^*) and B 2p(σ^*)? –

(b) The intensity of B 2p(π^*) spectra of the two kinds of B sample both increase at 188-192 eV regardless of the changes in test angle. So, does this mean that both cases exist the unoccupied out-of-plane B 2p(π^*). -

Strictly speaking, an ideal separation between π^ and σ^* resonances on the energy scale can be observed only in spatially confined atomic systems like small molecules. In itinerant solid-state systems, the electronic structure typically shows dispersing band that may have π and σ*

character in different symmetry points of the Brillouin zone. After integration over the Brillouin zone, the resulting DOS may retain π or σ character, but this definition is not very strict, as several bands may contribute to the same DOS feature. Nevertheless, the concept of π and σ states is a useful tool in describing electronic structure of certain 2D materials.

In our case, both B monolayers, whose XAS spectra are shown in Figure 3d-e, represent rather delocalized 2D systems where bonding is strongly affected by the underlying Ir substrate. As a result, π and σ bands can overlap and contribute to DOS features at the same energy. Therefore, to answer the reviewer's question, there is always a contribution from π^ states in the 188-192 eV energy region. The key difference between the two B monolayers is that for borophene on Ir (Figure 3d) we can clearly see π^* states at these energies, while for the buried B interlayer this region is obviously dominated by σ^* states (Figure 3e). For the latter case, the ground-state band-structure calculation (new Figure S8 in the SI) explains qualitatively the dominance of σ^* states in the B K-edge NEXAFS.*

11. The third row from the bottom on page 15. ".....monolayer of A atoms." Does the "A atoms" mean Au atoms? –

This is correct. We thank the reviewer for this remark and modify the text accordingly.

12. It is indeed a very interesting phenomenon that the pseudomorphic Au species will form a regular pattern of densely packed equilateral triangles after element B treatment. And the author does a lot of work to demonstrate the existence of the B layer between Au ML and Ir(111) and the truly Au 2D structure. And I believe in these conclusions. In addition, I am curious about why this happened. A scientific issue is why element B drives Au species to evolve into a new atomic layered pattern, and what is the role B plays in the process. If the authors can give more discussions or data to settle this issue, I believe this will bring more inspiration for the synthesis of 2D metallic materials. –

We thank the reviewer for the remark on the mechanism of Au nanopatterning, this is indeed a very interesting topic. In the revised manuscript, we have added a concluding paragraph in the section "Origin and nature of nano-structuring in Au mono- and bilayers" explaining our current understanding of this process:

"As a concluding remark, we would like to summarize our current understanding of the factors leading to the formation of nano-patterned Au mono- and bilayers upon embedding boron at the Au/Ir interface, a mechanism that may be relevant to other metal films. Ultrathin metal layers may become strongly strained on mismatched substrates (like Au on Ir), particularly at low temperatures (like room temperature). When B atoms arrive at such an interface at elevated temperatures, the system cools and gradually freezes introducing a strain field. This field dictates where B atoms can reside and where not, elevating those areas of the metal film where boron is present. For thicker metal films (more than 2 ML in the case of Au on Ir) the strain at the interface is considerably relieved, hampering the formation of strictly periodic nanostructures. As the strain field is specific to each adsorbate/substrate combination, we speculate that the nanopatterning motif can vary significantly for metals other than Au on the same B/Ir substrate."

As for the new data, we do have some very preliminary unpublished results where boron is seen to nanostructure metal films other than Au, and we do see interesting differences due to modified interfacial strain. However, this is a separate project that needs much more work to be completed.

13. For the reference part. According to the rules of Nature Communications, the format should be checked and revised carefully.

(a) For book citations. The publisher and city of publication are required.

(b) Article titles should be in Roman text, only the first word of the title should have an initial capital.

(c) Journal names are italicized and abbreviated (with full stops) according to common usage. In addition, please check the usage of the dot in the abbreviation of the journal name.

We are grateful to the reviewer for this observation and correct the references accordingly so that these would meet the required format.